# Induction of tunnelling nanotube-like structures by influenza A viruses requires the onset of apoptosis

Daniel Weir[1]*, Calum Bentley-Abbot[1], Jack McCowan[2], Colin Loney[1], Edward Roberts[2,3], Edward Hutchinson[1]*

1 MRC-University of Glasgow Centre for Virus Research, Glasgow, United Kingdom, 2 Cancer Research UK Scotland Institute, Glasgow, United Kingdom, 3 School of Cancer Sciences, University of Glasgow, Glasgow, United Kingdom

* d.weir.2@research.gla.ac.uk (DW); edward.hutchinson@glasgow.ac.uk (EH)

## Abstract

As well as spreading through virions, influenza A viruses (IAVs) can evade antiviral drugs and neutralising antibodies by spreading directly from cell to cell. In cell culture this can occur by the induction of intercellular membrane connections known as tunnelling nanotube-like structures (TLSs), which are capable of trafficking the viral genome between cells. Here, we showed that TLSs are formed by IAV infected cells *in vivo*, and then used *in vitro* models to ask how IAVs induce their formation. We found that TLS formation is not induced by cytokine signalling from infected to uninfected cells, but induction does require intracellular IAV replication. IAV replication can form filamentous virions which have structural similarities to TLSs, but we found that TLS induction is independent of virion morphology. We therefore looked at the intracellular responses to infection and found that the induction of TLSs correlated with the induction of apoptosis. Furthermore, the ability of IAVs to drive TLS formation can be modulated by chemically inhibiting, or inducing apoptosis. Finally, we found that inhibiting apoptosis, which prevents IAVs from inducing TLSs, lead to a significant reduction in the ability of IAVs to directly spread between cells. Our results, which suggest that IAVs can control their ability to spread directly from cell to cell by driving infected cells into apoptosis, identifies a new way in which a virus can manipulate its host to evade antiviral immune responses.

## Author summary

Influenza A viruses (IAVs) spread efficiently through the respiratory tract in the form of extracellular virus particles, but can be restricted by neutralising antibodies and antiviral drugs. IAVs can avoid this restriction by transporting viral genomes directly from one cell to the next. They can do this by inducing

**Data availability statement:** Underlying quantitative data is included within the attached file titled S1 Data. Raw data can be found at the following address: https://doi.org/10.5525/gla.researchdata.1966.

**Funding:** This work was supported by funding from the UK Medical Research Council (MRC), as a studentship to D.W [MC_ST_00034] and as core funding to C.L [MC_UU_12016/10] and Quinquennial funding to the MRC-University of Glasgow Centre for Virus Research [MC_UU_12014/9 and MC_UU_00034/7]. E.H. was funded by a Transition Support Award from the UK Medical Research Council [MR/V035789/1]. We also acknowledge funding from the Wellcome Trust as Four-Year PhD studentships in Basic Science to C.B.A [226861/Z/23/Z], and J.M [218518/Z/19/Z]. This work was also supported by Cancer Research UK (CRUK) core funding to the Scotland Institute [A31287] and to E.R. [A1920]. The funders had no role in study design, data collection and analysis, decision to publish, or preparation of the manuscript.

**Competing interests:** The authors have declared that no competing interests exist.

the formation of long, thin intercellular connections known as tunnelling nanotube-like structures, which are capable of trafficking viral genomes. In this study, we demonstrate for the first time that tunnelling-nanotube like structures form within IAV infected lungs. We then asked how IAVs induce these structures and found that cell death pathways triggered by IAV replication were required, and that blocking those pathways reduces the ability of IAV infection to spread directly between cells. Therefore, this host cell response to infection plays a critical role for establishing routes of infection spread between cells. In this way, the virus exploits the cell death response of its host to ensure that its infection can continue to spread even within the challenging environment of the respiratory tract.

## Introduction

Influenza A viruses (IAVs) require virions for transmission between host organisms. However, within an infected host IAV infection can spread between cells through alternative routes that are independent of extracellular virions [1–5]. This process, referred to as direct cell to cell spread, can ensure the continued intercellular transmission of IAV infection in the presence of neutralising antibodies and antiviral drugs [1–4,6].

Many viruses can undergo direct cell to cell spread and there are diverse mechanisms by which they do so (reviewed in [6]). Some of these transport complete virions directly from one cell to another without releasing them into the extracellular environment. For example, filopodial bridges and virological synapses can deliver newly formed virions of alphaviruses and human immunodeficiency virus (HIV), respectively, to adjacent cells through cellular junctions or synaptic clefts [7–9]. Other mechanisms use membrane fusion to enable the direct transfer of viral genomes and proteins. This can be seen in the formation of syncytia from directly adjacent cells [10], as well as in connections made between distant cells through open ended, long-range connections known as tunnelling nanotubes (TNTs) [11]. TNTs are thin (typically 50–200 nm in diameter), F-actin rich cell connections that function in trafficking various cargos [12,13] and have been implicated in the spread of several viral pathogens [11] including herpes simplex virus (HSV), HIV, SARS-CoV-2 and IAVs [1,14–17].

Under a strict definition, TNTs can be distinguished from other cellular projections by their cytoskeletal content (F-actin), their ability to connect cells over a long distance, and their ability to traffic cargo between them [12]. In addition, unlike other forms of direct cell connections, TNTs can extend from cells and be suspended above the substratum [12]. However, due to the challenges demonstrating all of the definitive properties of TNTs when classifying these fragile and transient cell connecting structures [12], it is often impractical to apply each of these criteria when analysing large quantities of imaging data. Therefore, intercellular connections that meet most but not all of these criteria are typically referred to as TNT-like structures (TLSs)

[18–21]. In this current study, we classified F-actin containing structures connecting cells over a minimum distance of 5 μm as TNT-like, and we excluded those that have likely resulted from recent cell division or death (e.g., structures with cellular midbodies, or involving cells with fragmented nuclei).

Studies of *in vitro* models have demonstrated the potential importance of TLSs in IAV replication. IAV infection induces the formation of TLSs, creating an F-actin- and Rab11a-dependent mechanism for the direct transmission of IAV between cells and promoting the reassortment of IAV genome segments [1,3,5]. However, it was unclear if TLSs could actually form in the dense tissues which IAV naturally infects, as TLSs are delicate and detecting them within respiratory epithelia is extremely challenging. In addition, it was not known how IAV infection of cells was able to induce TLS formation.

In this study, we addressed these questions. Using a reporter mouse system, we observed TLSs forming from IAV infected cells within the lung epithelium, supporting the relevance of TLS trafficking during natural IAV infections. Using *in vitro* assays, we then asked how IAV infection induces TLS formation. Having ruled out cytokine signalling from infected to uninfected cells, as well as processes linked to the formation of virions, we found that the induction of TLSs by IAVs results from the triggering of apoptosis. Outside the context of infection, the onset of apoptosis is known to drive TLS formation between healthy and apoptotic cells [22,23]. We found that TLS induction by IAV correlated with the triggering of apoptosis, and then we show that a pan-caspase inhibitor prevented the ability of IAV infected cells to induce the formation of TLSs. Conversely, whilst the apoptosis-inducing drug cisplatin did not trigger TLS formation by itself, when combined with IAV infection, cisplatin promoted TLS formation at an earlier timepoint in infection. Furthermore, inhibiting TLS formation by blocking apoptosis reduced the direct cell to cell spread of IAV infection between spatially separated cells, supporting a functional role of apoptosis in facilitating viral genome transfer through TLSs. Our data therefore reveal that TLS connections between cells are a feature of natural IAV infection and their induction is driven by IAV replication triggering the onset of apoptosis.

## Results

### TNT-like structures are present within the lungs of IAV infected mice

To date, the study of TLSs and their consequences for IAV infection has been limited to *in vitro* models. However, for TLSs to be relevant to the intercellular spread of IAV infections they need to be present at the site of natural infection. Within the lung, TLSs have been observed within solid tumours [24] but not within typical lung epithelium. Conceptually, the lung epithelium is a challenging environment for cells to form thin, long-range intercellular projections. These already fragile and transient structures would need to be able to form in a cell dense and complex environment whilst withstanding the mechanical deformations during breathing, as well as either penetrating solid tissue or surviving mucociliary flow. Furthermore, the visualisation of TLSs in tissue, either *in vivo* or *ex vivo,* has been mostly limited to tissues with lower cell densities and/or higher rigidities than the respiratory epithelium [14,25–27], and only some non-infection based stress or pathologies have been studied (e.g., lipopolysaccharide treatment [25], or within solid tumours [27–29]). Furthermore, most imaging of TLSs has been achieved using methods with minimal staining and tissue manipulation [24,28,29], and the study of TLS formation between epithelial cells following respiratory virus infection is challenged by the need to distinguish between infected and healthy cells.

To assess the ability of TLSs to form at the site of a respiratory infection, we needed a method which uniquely labelled infected cell membranes with minimal tissue manipulation to maintain TLS integrity prior to imaging. We did this using the mT/mG reporter mouse system in combination with an IAV (A/Puerto Rico/8/1934 H1N1, referred to as PR8) which has been genetically modified to encode the Cre recombinase (PR8 Cre) [30]. The mice encode a membrane-targeted tdTomato (mT) fluorophore flanked by *loxP* sites, such that when Cre recombinase is expressed, the *tdTomato* gene is replaced by a downstream membrane-targeted *GFP* gene (mG, Fig 1A) [30]. The result is that infection with a Cre-expressing virus permanently changes the membrane fluorescence of a cell. Accordingly, we infected mT/mG mice with PR8-Cre and, six days post infection, we harvested the lungs. Infected cells were identified by mG positive

PLOS Pathogens

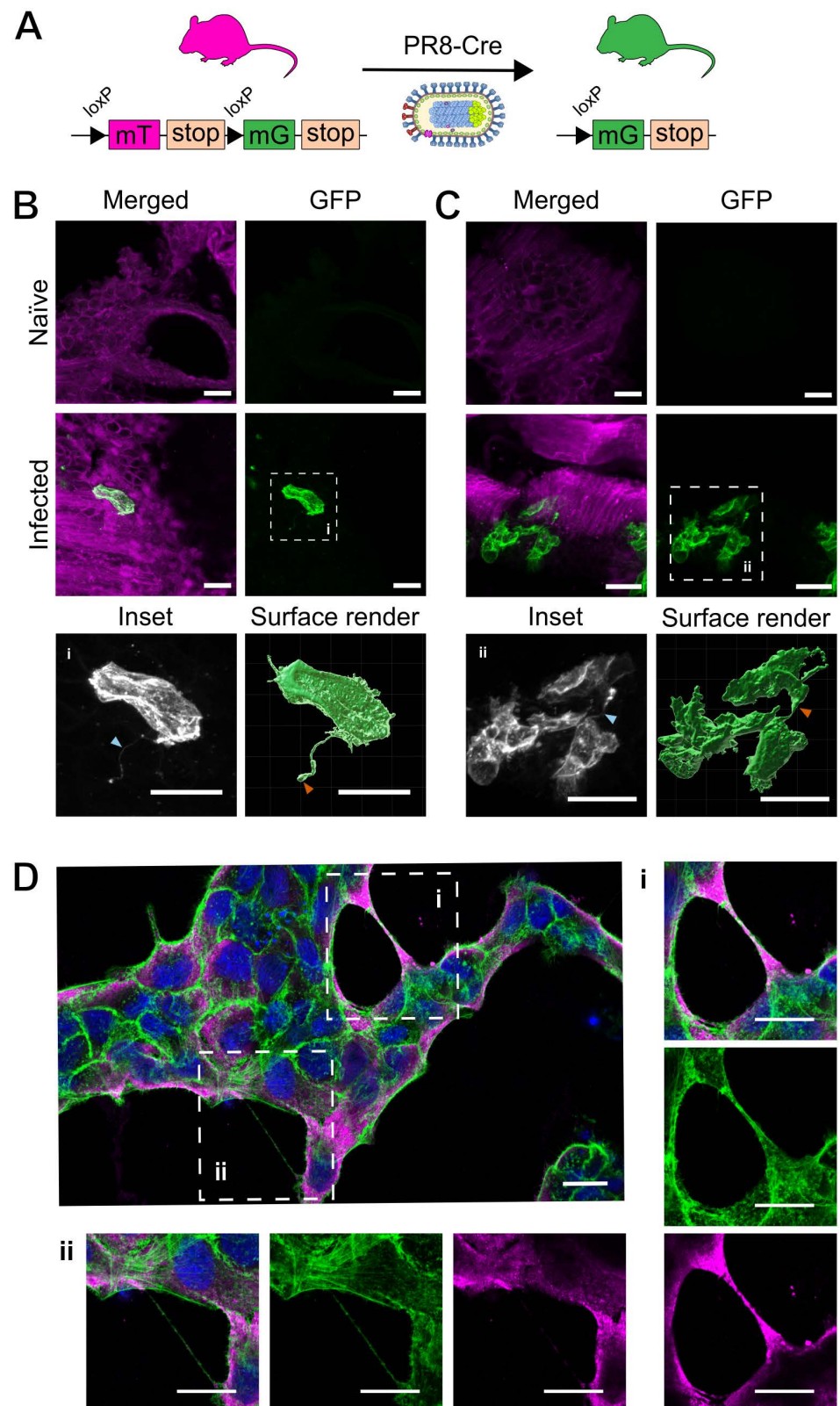

**Fig 1. TNT-like structures involving IAV infected cells are observed *in vivo* and *in vitro*. (A)** Schematic of mT/mG reporter mouse system in combination with an IAV encoding for Cre (PR8-Cre), with arrows indicating *loxP* sites that flank the tdTomato fluorophore. Following intranasal infection

and Cre recombinase expression, the membrane targeted tdTomto (mT) is replaced with a membrane targeted GFP (mG) fluorophore. Six days after intranasal infection with PR8-Cre, lungs were harvested from mT/mG mice and thick sections were prepared for confocal microscopy. **(B & C)** Maximum intensity projections of TNT-like structures (TLSs) within lung tissue **(B)** projecting from an isolated PR8-Cre infected cell (distinguished by the expression of GFP), and **(C)** connecting infected cells. Magnified insets outlined in white boxes are shown (i-ii) alongside surface renders. Membrane labels are tdTomato (magenta) and GFP (green/white). Blue arrowheads indicate the presence of TLSs, and orange arrowheads indicate the bulbous termini of TLSs. **(D)** Tiled image of a single field of view (63x magnification), representative of images used for TLS quantification, showing TLSs of varying thickness between MDCK cells at 16 hours post infection with PR8 at an MOI of 1.5 PFU/cell. Nuclei (blue), F-actin (green), nucleoprotein (magenta). All scale bars = 20 µm.

membrane fluorescence, and in order to capture cellular projections of various lengths and orientations within the tissue, thick tissue sections were imaged using super resolution confocal microscopy, incorporating Z-stacks to facilitate 3D rendering of structures and maximum intensity projections (Fig 1B and 1C).

Within the lungs of inoculated mice, we observed individual IAV infected epithelial cells in the lower respiratory tract. These cells have the appearance of pseudostratified columnar epithelial cells, which are known to support IAV infection [31,32]. From the surface of these cells we could observe the protrusion of thin TLSs (Fig 1B). Interestingly, these appeared to have formed in the spaces between cells of the epithelium rather than through the lumen of the airway, resulting in a curved structure that followed the contours of neighbouring cells. In addition, TLSs seemingly connecting infected cells were seen (Figs 1C and S1), suggesting that TLSs might be capable of transferring IAV infection within a tissue, or of facilitating the interaction of separately infected cells. The structures we observed in the airway epithelium have similar characteristics to those previously seen in mammalian tissues such as the cornea of mice, in which TLSs are often curved and have a bulbous structure at its terminus (Fig 1B) [25]. We noted that the structures we observed were shorter than those seen in some other mammalian tissues, such as solid lung tumours [24]. This would be compatible with the previously posed hypothesis that TLS length is limited in tissues with higher cell densities (longer TLSs are reported in the centre of the cornea when compared with the periphery of this tissue, the latter of which has a higher density of cells) [25]. To the best of our knowledge, this is the first demonstration of TLSs within infected but otherwise normally structured lung tissue, and the first demonstration of these structures being able to form within a layer of epithelium. Quantitation of TLS formation *in vivo* was not possible, due to high levels of variability in both the tissue structure and the distribution of IAV infection across tissue sections, as well as the likelihood of damage to these fragile structures if samples underwent additional staining. Nevertheless, the presence of a number of TLSs within a tissue that undergoes repetitive expansion and deflation suggests a surprising degree of robustness compared to the properties of these structures *in vitro* [12].

Having demonstrated that TLSs do form at the natural site of infection, we needed an experimental system which was easier to manipulate and control, and which was more amenable to high throughput imaging, in order to study how IAV infection triggers the formation of TLSs. We therefore turned our attention to *in vitro* infections, and applied an optimised fixation method [33] to IAV infected cells. This allowed us to preserve TLSs with a variety of thicknesses, and presumed stabilities, for imaging (Fig 1D). Consistent with previous cell culture based studies, we found that infected cells typically produce straight TLSs that contain punctae of viral nucleoprotein (NP), indicating the incorporation and transport of the viral ribonucleoprotein (vRNP) complex (Fig 1D) [1,3,5]. This demonstrated the suitability of our *in vitro* approach for studying how IAV infection induces TLS formation.

### TNT-like structures are formed by both infected and uninfected cells, with no evidence of pathfinding

Whilst the mT/mG reporter mouse system revealed that TLSs were able to extend from infected epithelial cells *in vivo*, the relative tendency of these structures to originate from or be received by infected and uninfected cells was unknown. A preference for either infected or uninfected cells would suggest TLS pathfinding, a process by which TLSs can be guided from a TLS donor cell along a concentration gradient of a secreted factor produced by the TLS recipient cell [34]. TLS

pathfinding has been demonstrated to occur between homotypic astrocyte cultures and cocultures with HEK293 cells, along a concentration gradient of the small extracellular protein S100A4 [35]. As TLS pathfinding determines which cells form TLS connections it can have functional consequences. For example, diseased cells lacking functional lysosomes utilise TLSs to facilitate the transfer of lysosomes from healthy cells [36,37].

In the context of virus infection, evidence of TLS pathfinding varies and appears to be virus-specific. For example, TLSs have been reported to originate at comparable rates from HTLV-1 positive and negative cells in a co-culture [38]. In other cases, directional TLS formation has been observed, for example during vaccinia virus infection – though in this case it was driven by vaccinia virus proteins on cell surfaces which, upon interacting with secondary incoming virions, trigger the growth of TLSs towards uninfected cells [39]. Here, we asked whether TLS pathfinding would occur during IAV infections, as this would promote the cell to cell spread of IAV.

IAV infection results in the production of a wide variety of extracellular signalling molecules, such as interferon (IFN; reviewed in [40,41]), that can trigger an antiviral response in neighbouring cells. We hypothesised that a potential TLS chemoattractant could be amongst the paracrine innate immune signals produced by IAV infected cells. If this was the case, we would expect to observe TLSs preferentially forming from uninfected to infected cells. Previous research by Kumar *et al.* showed that IAV infected and uninfected A549 cells (a common *in vitro* model for the study of lung epithelial cell infection, which can mount an innate immune response [42]), can be involved in TLS connections, with both infected and uninfected cells being able to produce TLSs [3]. To quantify how TLS origination or receipt varied according to IAV infection status, we modified A549 cells to constitutively express a membrane targeted AcGFP fluorophore [38,43]. Modified A549 cells retain the membrane label well during sequential passaging, and when cocultured with WT A549s, the membrane fluorescence of a TLS allowed the cells which originated the structure to be clearly identified. We then infected the co-culture at a low MOI and determined the recipient and donor cells of the TLSs that formed (Fig 2A). We first observed that modified A549s could be infected by IAV and form TLSs at rates comparable to the WT A549s (S2B Fig). Under these low MOI conditions, the presence of an IAV infection within a subset of cells is expected to exert an influence on all cells through the production of extracellular signalling molecules. We then calculated the proportion of uninfected and infected cells producing a TLS and found no significant difference between them (Fig 2B). Infection status of individual cells also did not influence the mean TLS length (approximately 9700 nm; Fig 2C). Therefore, under the low MOI infection conditions of this assay, where the minority of cells were infected (Fig 2D), the infection status of individual A549 cells did not determine their ability to initiate a TLS or the length over which interactions could form.

These observations suggested that the likelihood of uninfected and infected cells being involved in a TLS connection would be proportional to their abundance. According to NP staining, 37% of cells in the co-culture were infected and 63% uninfected (Fig 2D). This was reflected in the proportion of cells involved in a TLS that were infected (Fig 2E), which suggests that TLSs were not preferentially originating from or contacting infected or uninfected cells. This is further supported by the observation that a similar percentage of total uninfected and infected cells were involved in a TLS (Fig 2F). We then focused on TLS connections between heterotypic cell pairs (WT and AcGFP labelled) which displayed an asymmetric infection status (i.e., infected and uninfected), as this allowed us to correlate TLS directionality to the infection status of donor and recipient cells. We observed that approximately 70% of TLSs mediating these cell connections originated from the uninfected cell (Fig 2G and 2H), which once again was consistent with the proportion of uninfected cells in the assay (Fig 2D). Together, these data indicate that under these low MOI infection conditions *in vitro*, TLS pathfinding is not observed between individual cells displaying an asymmetric infection. Therefore, the extracellular signals produced by an IAV infected cell are seemingly not functioning as TLS chemoattractants.

## IAVs induce TNT-like structures through viral replication but not by the production of secreted factors

Multiple studies have reported the induction of TLSs following high MOI IAV infections [1,3,5], but the reason for this was unknown. We therefore set out to determine how this induction occurs. The absence of TLS pathfinding suggested that

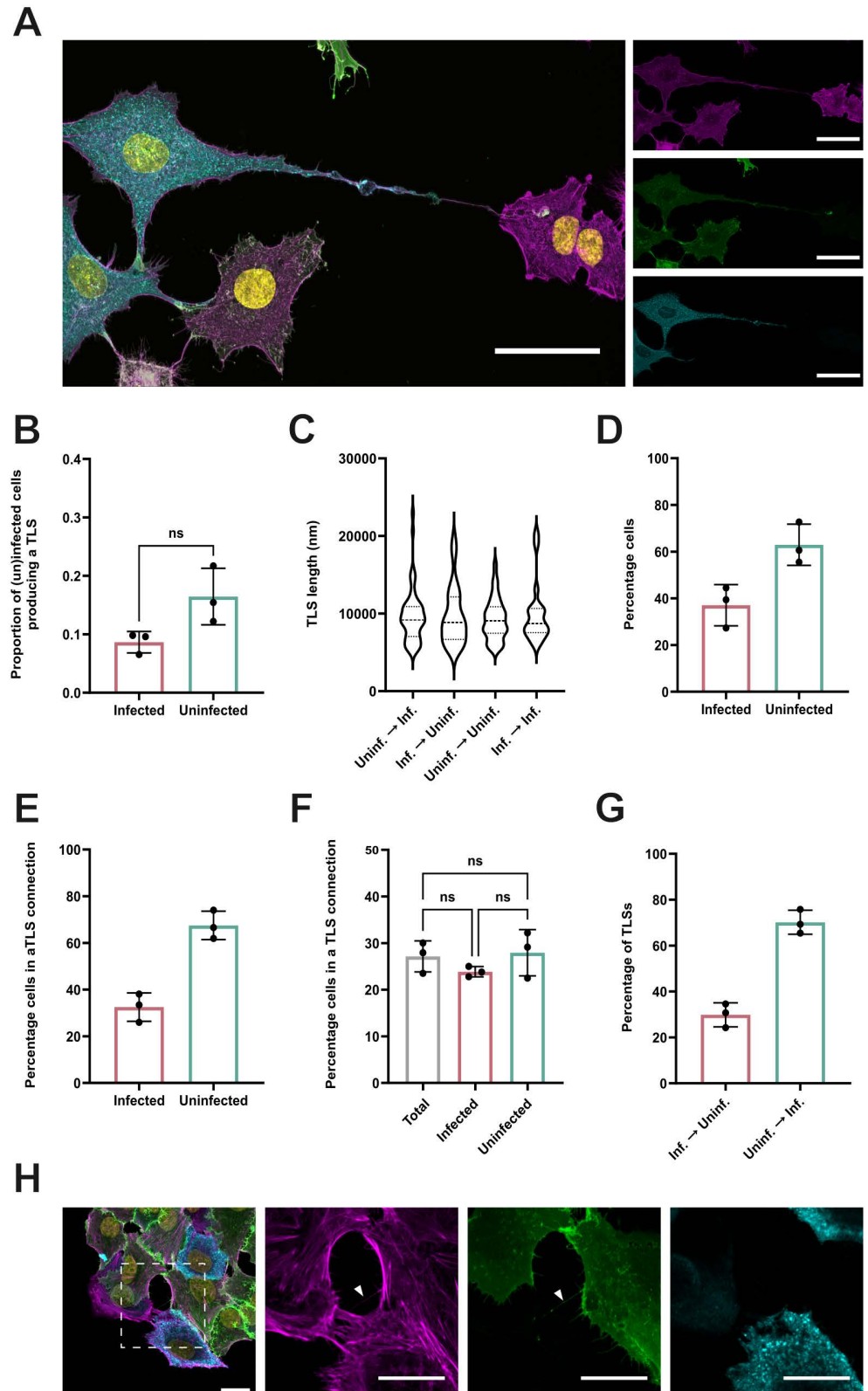

**Fig 2. Infected and uninfected A549 cells both initiate and are contacted by TLSs at similar frequencies in co-cultures. (A)** A TLS structure forming from a PR8 infected, AcGFP membrane labelled A549 cell, connecting to an uninfected, WT A549 cell. Nuclei (yellow), F-actin (magenta), AcGFP

(green), NP (blue). Scale bar = 50 μm. **(B)** The proportion of uninfected and infected cells producing a TLS in a low MOI infection. The difference between the proportions of uninfected and infected cells that produced a TLS was tested for significance by Mann-Whitney test (n.s. $p > 0.05$). **(C)** The length of TLSs connecting cells with either an asymmetric or symmetric infection status, shown as violin plots with the median and the upper and lower quartile values indicated by dashed lines. Differences in mean TLS length was tested for significance by Kruskal-Wallis test (n.s. $p > 0.05$). **(D)** The percentage of co-cultured cells that were infected or uninfected and **(E)** the percentage of cells involved in a TLS connection that were either uninfected or infected. **(F)** The percentage of total cells, total infected cells and total uninfected cells involved in a TLS connection. Differences between infection status were tested for significance by Kruskal-Wallis test (n.s. $p > 0.05$). **(G)** Percentage of TLSs originating from an infected or uninfected cell, collected from connected pairs of WT and AcGFP membrane labelled A549 cells (i.e., heterotypic cell pairs) with an asymmetric infection status. For all data, the mean and SD is shown (n = 3). **(H)** Representative confocal image of a heterotypic cell pair displaying asymmetric infection status, connected by a TLS originating from the uninfected cell. Nuclei (yellow), F-actin (magenta), AcGFP (green), NP (blue). Scale bar = 20 μm.

TLS formation is not regulated by factors secreted by the cells in response to IAV infection. Conversely, outside the context of infection, secreted factors have been shown to cause TLS induction. IFN-α treatment has been shown to drive TLS induction within Kcl-22 cells [43], and conditioned media from macrophages has been shown to induce TNTs within the breast cancer cell line MCF-7, potentially by the secretion of paracrine cytokines and chemokines [44,45]. We therefore decided to test whether paracrine signals cause TLS induction in the context of a high MOI IAV infection.

To examine this, we used MDCK cells due to their high permissiveness to IAV infection, their ability to produce innate immune signals, and their previous use for the study of TLS induction [1,46]. We collected the medium from MDCK cells 16 hours after infection (by which point abundant viral replication has occurred and TLSs can be readily observed, Fig 1D) using the influenza strain A/Udorn/307/1972 H3N2 (referred to as Udorn, a lab adapted human IAV strain first shown to induce TLSs, [1]). We used U.V. irradiation to inactivate the virions in this medium before overlaying it onto fresh cells (Fig 3A). Sixteen hours later, we used immunofluorescence, both to confirm virus inactivation and to quantify the number of TLSs that had formed (Fig 3Ai). Cells overlaid with U.V.-treated supernatant lacked any viral NP expression, confirming that virus inactivation had been effective (Fig 3Bi and 3Bii). In these cells there was also a lack of TLS induction, with TLS formation comparable to that in cells treated with media from mock infections (Fig 3C). TLS induction only occurred when additional, replication-competent virus was added to the cells (Fig 3Biii and 3C). To test whether U.V. treatment could have inactivated innate immune signals in the conditioned medium, we harvested cells which had been exposed to the conditioned medium and used immunoblotting to detect phosphorylated STAT1 (pSTAT1), a signalling molecule phosphorylated during type 1 IFN signalling, in the cell lysates (Fig 3Aii). We observed that pSTAT1 abundance doubled in cells overlaid with U.V. treated conditioned media taken from infected cells when compared to media taken from mock infected cells (Fig 3D and 3E). This demonstrates that U.V. treatment did not prevent the conditioned medium from carrying functional innate immune signalling molecules, and therefore shows that these molecules were not responsible for TLS induction.

To further examine the hypothesis that TLS induction was not caused by innate immune signalling, we treated MDCK cells with 2 μM ruxolitinib, a broad-spectrum Janus kinase (JAK) inhibitor which prevents the phosphorylation of STAT1 (Fig 3F). Despite the inhibition of interferon signalling, IAV infection still clearly induced the formation of TLSs (Fig 3G and 3H), at levels comparable to those previously seen (Fig 3C). Together these results strongly suggest that TLS induction during IAV infection is independent of secreted extracellular factors, including the production of cytokines that signal through the JAK/STAT1 pathway.

### IAVs induce TNT-like structures and undergo direct cell to cell spread consistently across strains of differing virion morphologies

As the induction of TLSs was dependent on viral replication, we reasoned that it might be caused by some feature of the viral replication cycle, and we looked to see if differences in the level of TLS induction between strains of IAV could help to understand this. In particular, we noted a striking similarity between the filamentous virions formed by some IAVs and the physical features of TLSs.

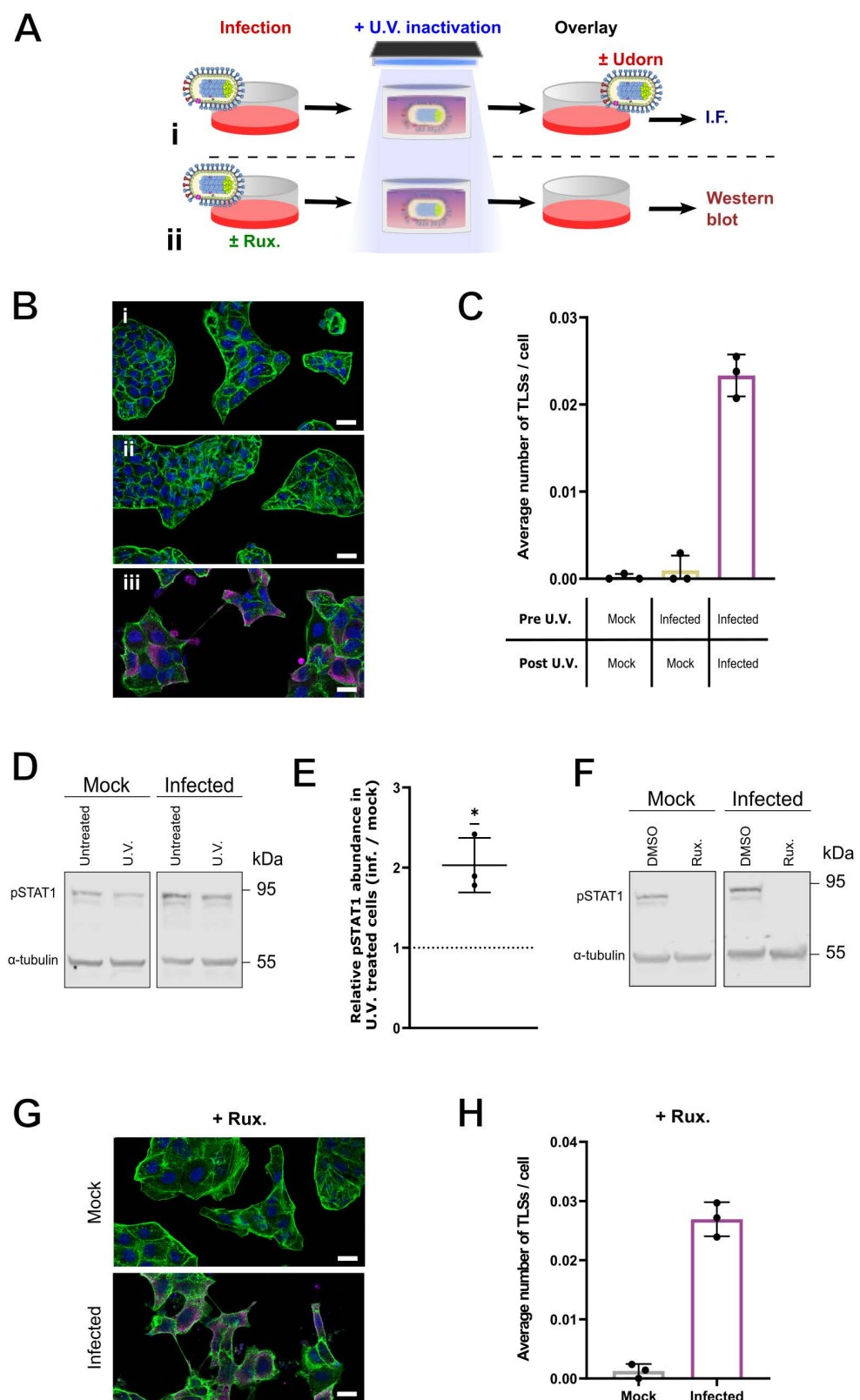

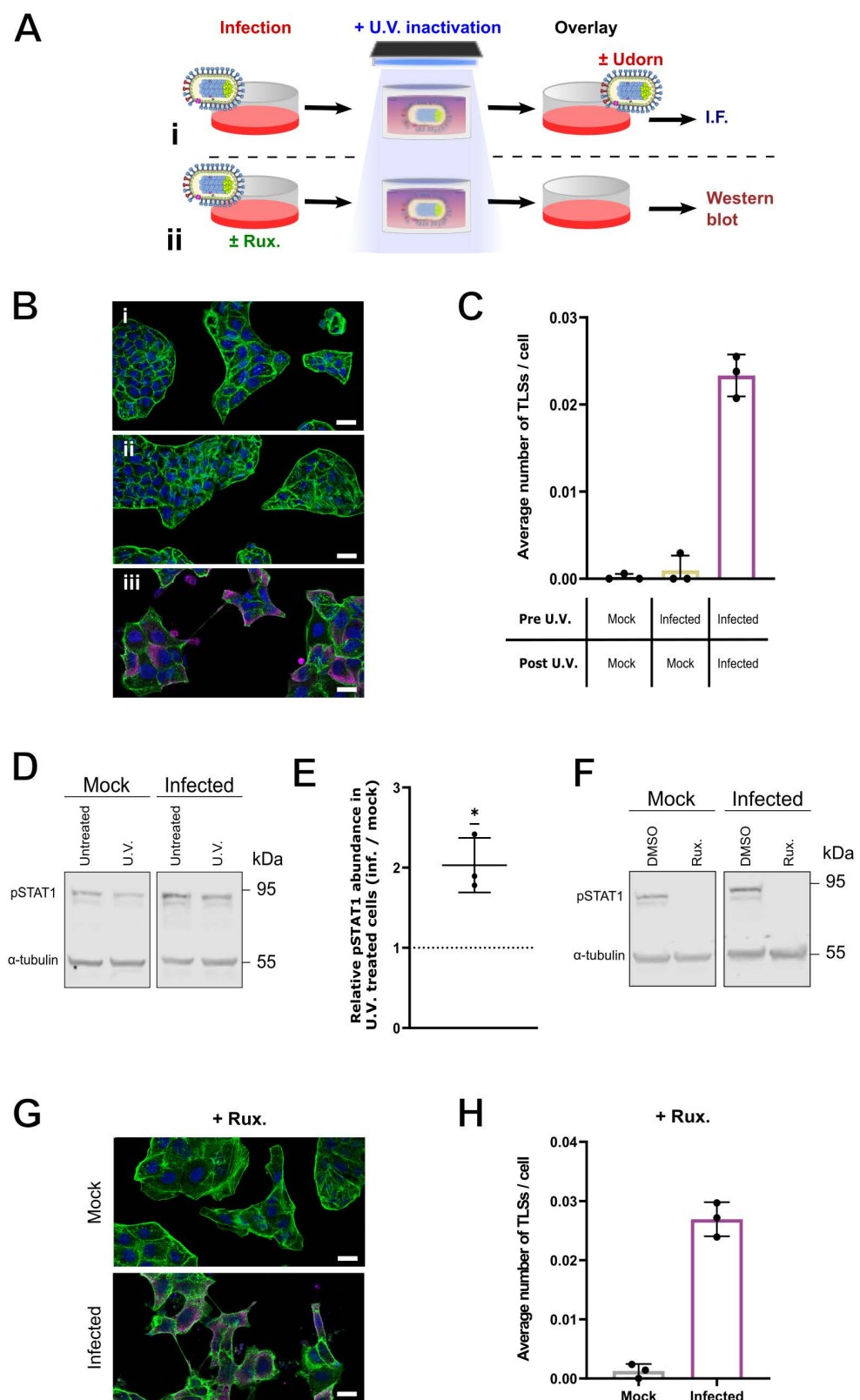

**Fig 3. Replicating IAVs, not paracrine signals, are required for TLS induction. (A)** Schematic of experimental design, showing the U.V. treatment of conditioned media collected from either mock or Udorn infected cells at 16 hours post infection, overlaid onto fresh cells for either downstream **(i)**

immunofluorescent (I.F.) staining and confocal imaging or **(ii)** cell lysate harvesting and western blotting. **(B)** Representative confocal images of cells treated with the following overlays: **(i)** U.V. treated supernatant from mock infected cells, **(ii)** U.V. treated supernatant from infected cells and **(iii)** U.V. treated supernatant from infected cells, and additional Udorn virus. Nuclei, (blue), F-actin (green), NP (magenta). Scale bar = 20 μm. **(C)** Average number of TLSs per MDCK cell 16 hours following treatment with the overlay media described. **(D)** Western Blots for pSTAT1 using cell lysates harvested 16 hours post treatment with U.V. inactivated conditioned media from either mock infection or Udorn infection. Images are representative of three biological repeats. **(E)** Ratio of the relative abundances of pSTAT1 at 16 hours post treatment with U.V. inactivated supernatants from infection, normalised to mock infection. No change is indicated by a dashed line; the significance of the difference from this was tested by a one-sample t-test (*$p < 0.05$). **(F)** Western blot of pSTAT1 from cells infected with Udorn at an MOI of 1.5 PFU/cell with the addition of either DMSO or 2 μM ruxolitinib (Rux.) to the SFM overlay, with cell lysates being harvested 16 hours post infection. **(G)** Representative confocal images of MDCK cells at 16 hours post infection. with Udorn at an MOI of 1.5 PFU/cell in the presence of 2 μM Rux. Nuclei (blue), F-actin (green), NP (magenta). Scale bar = 20 μm. **(H)** Average number of TLSs per cell in the presence of Rux. following mock or Udorn infection. For all data the mean and SD are shown (n = 3).

IAV virions are pleiomorphic, with morphologies ranging from spherical to filamentous [47,48]. Although the most widely-used laboratory strains of IAVs have lost their ability to form filamentous virions, clinical and veterinary isolates typically produce a variety of virions including long filaments (reviewed in [49]), and natural infections seemingly select for filamentous particles [50–52]. This suggests that the ability to form filaments might provide an advantage within the host. In support of this idea, it has recently been shown that filamentous virions can enhance infectivity and fusion even in the presence of neutralising antibodies [53]. It has also been proposed that the filamentous morphology may enhance direct cell to cell spread of IAV infection [48], which could confer a neutralising antibody evading route of infection spread. We observed that filamentous IAV virions and TLSs are comparable in their dimensions, composition, and in the cellular processes involved during their formation [34,49,54]. For these reasons, we hypothesised that the ability to form filamentous virions might correlate with the ability of IAV to induce TLSs.

To test this, we used PR8 and Udorn viruses, which are known to retain a predominately spherical or filamentous virion morphology, respectively [55,56]. By swapping segment seven of the viral genome, which encodes the matrix protein and with it the primary determinants of virion morphology, between these viruses, we generated reassortants of Udorn with the matrix of PR8 (Udorn MPR8) and PR8 with the matrix of Udorn (PR8 MUd). These segment seven reassortants are known to display intermediate virion morphologies compared to the wild-type (WT) viruses [57,58]. To measure virion morphology we used immunostaining of haemagglutinin (HA), both on infected MDCK cell membranes during viral budding (S4 Fig), and on free virions fixed on to glass slides (Fig 4A). By quantifying the lengths of filamentous particles within freshly harvested supernatants 48 hours post infection, we confirmed the predominately spherical and filamentous morphologies of PR8 and Udorn, as well as the intermediate morphologies of Udorn MPR8 and PR8 MUd (Fig 4B).

We then investigated whether the extent of TLS formation differed between these IAVs. To do this, we first infected subconfluent MDCKs with WT PR8 or Udorn at an MOI of 1.5 PFU/cell. At 16 hours post infection the cells were fixed, stained, and TLSs quantified from super resolution confocal micrographs (Fig 4C). We found that both viruses induced the formation of TLSs when compared to mock infected cells (Fig 4D), and that many of these TLSs contained NP (Fig 4E), suggesting that they could transport viral genomes. However, we found no statistically significant difference in the degree of TLS induction between the two viruses (Fig 4D). Indeed, the greatest number of TLSs per cell (though this was not significantly different) was induced by the spherical virus, PR8. This suggests that TLS induction is a common feature of IAV infection, and that the ability of a virus to produce filaments with a length greater than approximately 12 μm does not affect TLS formation. To confirm this, we tested the predominately spherical Udorn MPR8 virus and found that the levels of TLS induction were not elevated when compared to filamentous WT Udorn (Fig 4D).

Next, we investigated whether differences in virion morphology influenced the frequency of IAV direct cell to cell spread. To do this we implemented a microplaque assay, similar to that used previously by Roberts *et al.* [1]. In this assay, a confluent monolayer of MDCKs is infected at a low MOI and incubated under conditions inhibitory to the spread of infection by virions. This included the absence of TPCK trypsin (a protease that cleaves the viral receptor binding protein HA, priming

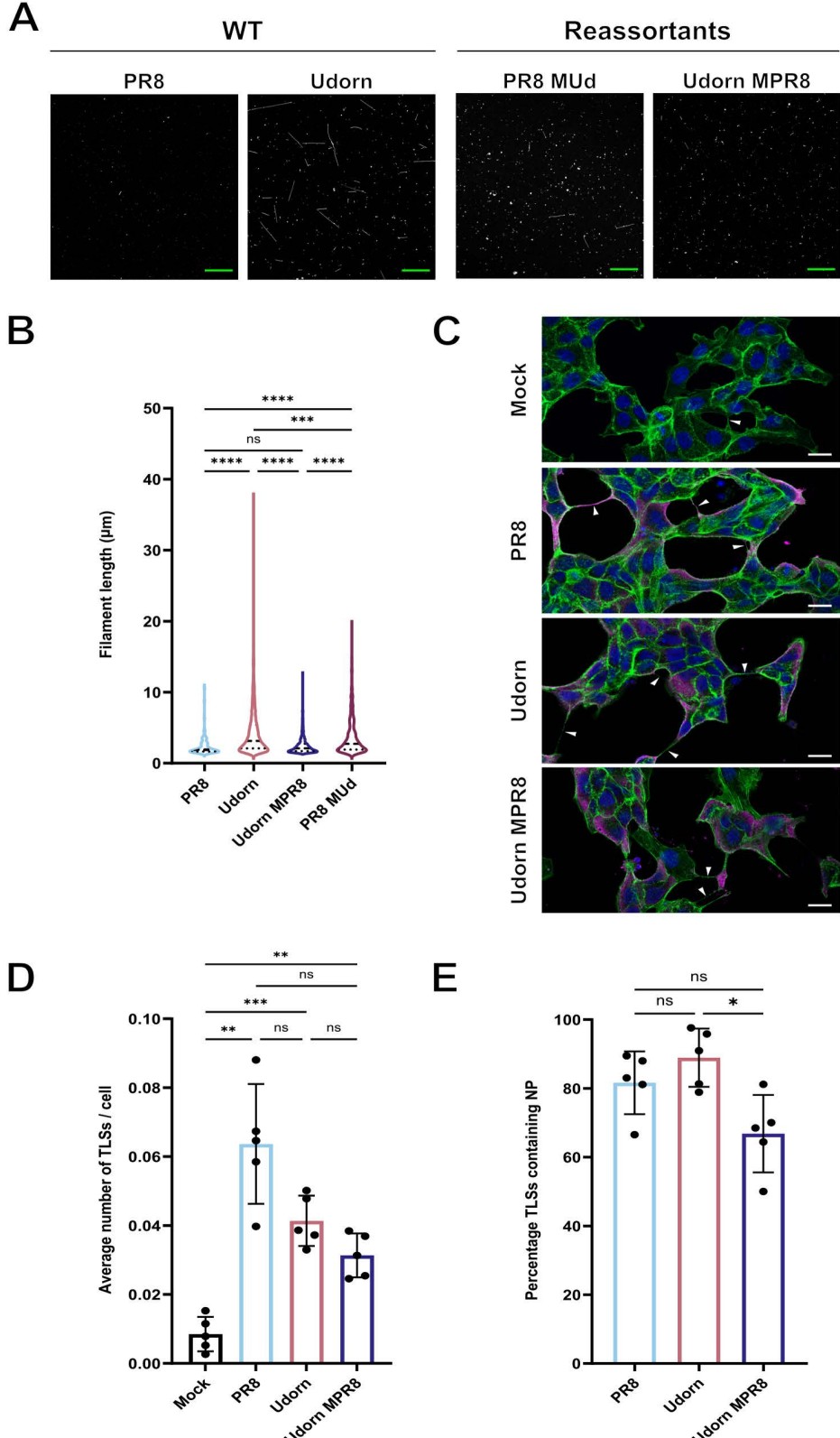

**Fig 4. TLS induction is not influenced by IAV virion morphology. (A)** Representative immunofluorescence images of influenza virions, harvested and fixed on coverslips at 48 hours post infection and labelled for hemagglutinin (HA) (white). Scale bars (green) = 20 μm. **(B)** Violin plot of the

filament lengths formed by different IAV strains, with the median and the upper and lower quartile values indicated by dashed lines. Individual filament lengths across three independent experiments were plotted. Differences in the mean filament length between strains was tested for significance by Kruskal-Wallis test (n.s. $p > 0.05$, ***$p < 0.001$, **** $p < 0.0001$). **(C)** Representative confocal images of MDCK cells at 16 hours after mock, PR8, Udorn or Udorn MPR8 infection. White arrowheads indicate the presence of TLSs. Nuclei (blue), F-actin (green), NP (magenta). Scale bars = 20 µm. For each virus, confocal images of infected cells were used to determine **(D)** the average number of TLSs per cell and **(E)** the percentage of TLSs containing NP. Differences between strains in TLS induction and in the incorporation of NP were tested for significance by one-way ANOVA (n.s. $p > 0.05$, *$p < 0.05$, **$p < 0.01$, ***$p < 0.001$). The mean and SD from 5 independent experiments is shown.

the virus for entry), and the inclusion of the neuraminidase inhibitor (NAi) zanamivir, which prevents the release of newly formed virions from infected cells. Under these conditions, the spread of infection to neighbouring cells should only result from direct cell to cell spread rather than from the release of cell free virions. The spread of infection to neighbouring cells can then be detected as clusters of adjacent NP-positive cells, or microplaques (Fig 5A).

To validate that PR8 and Udorn virion mediated spread is being inhibited by these conditions, we first assessed their dependency on TPCK trypsin. To do this we determined the ability of these viruses to cause cytopathic effect (CPE), as an indicator of large scale multicycle replication, either in the presence or absence of TPCK trypsin. We demonstrate that both PR8 and Udorn required TPCK trypsin for CPE to be observed (S5A Fig), indicating that infection spread of these viruses was greatly reduced when the exogenous protease is excluded and virion entry is not supported.

Next, we tested the effects of zanamivir on inhibiting virion release. We found that CPE could also be prevented by the addition of zanamivir even at low concentrations and in the presence of TPCK trypsin (S5A Fig). To confirm that the release of infectious virions was robustly inhibited by zanamivir, we infected cells with PR8 and Udorn in the presence or absence of 0.36 mM zanamivir, and without TPCK trypsin (i.e., both conditions of the microplaque assay). We collected the supernatants 48 hours post infection and then incubated them with TPCK trypsin to cleave and activate any released virions. We then used a plaque assay to detect any infectious virions, rinsing cells after adsorption to remove residual zanamivir and adding TPCK trypsin to the overlay (Fig 5B). Near-complete cell loss occurred with media collected from cells infected without zanamivir (Fig 5B), while no plaques were seen with media collected from cells infected in the presence of 0.36 mM zanamivir (Fig 5B), indicating that this antiviral drug completely blocked the release of infectious virions. Furthermore, when we look at the effect of zanamivir on infection spread between individual cells, the inclusion of zanamivir, even at concentrations less than 0.36 mM, resulted in an immediate reduction in the frequency and scale of microplaques, as expected (S5B and S5D Fig), with both viruses having similar sensitivities to zanamivir treatment (S5C Fig). Therefore, our experimental system that prevents progeny virion release and entry by including zanamivir and excluding TPCK trypsin respectively, provides a robust system to study the direct cell to cell spread of PR8 and Udorn viruses.

Having established that the conditions of the microplaque assay prevented infection through cell free virions, we compared the frequency of direct cell to cell spread of IAVs of differing morphologies by quantifying the size and number of microplaques that formed 48 hours post infection. We found that the IAVs tested were all capable of direct cell to cell spread, with between 25 and 40% of infected cells spreading infection directly to their neighbours (Fig 5C). The ability to form abundant filamentous virions of lengths greater than 12 µm did not provide any advantage for the cell to cell spread of IAV infection in this system (Fig 5C). Indeed, the spherical PR8 strain was slightly more effective at infecting neighbouring cells than the filamentous Udorn strain ($p = 0.0134$, Kruskal-Wallis test) (Fig 5C). This difference was not seen between the segment 7 reassortant viruses, strongly suggesting that the difference in direct cell to cell spread between the WT viruses is not primarily conferred by the morphology of the budding virion (Fig 5C). Furthermore, the eventual size of microplaques by 48 hours post infection was not influenced by strain tested (Fig 5D). Together, this indicates that the ability of IAVs to undergo direct cell to cell spread is not determined by the ability of an IAV to produce abundant, lengthy filamentous virions.

We hypothesised that the difference in the efficiency of microplaque formation between PR8 and Udorn (Fig 5C) could result from a strain dependent preference for the mechanism of direct cell to cell spread. Currently, two broad parallel

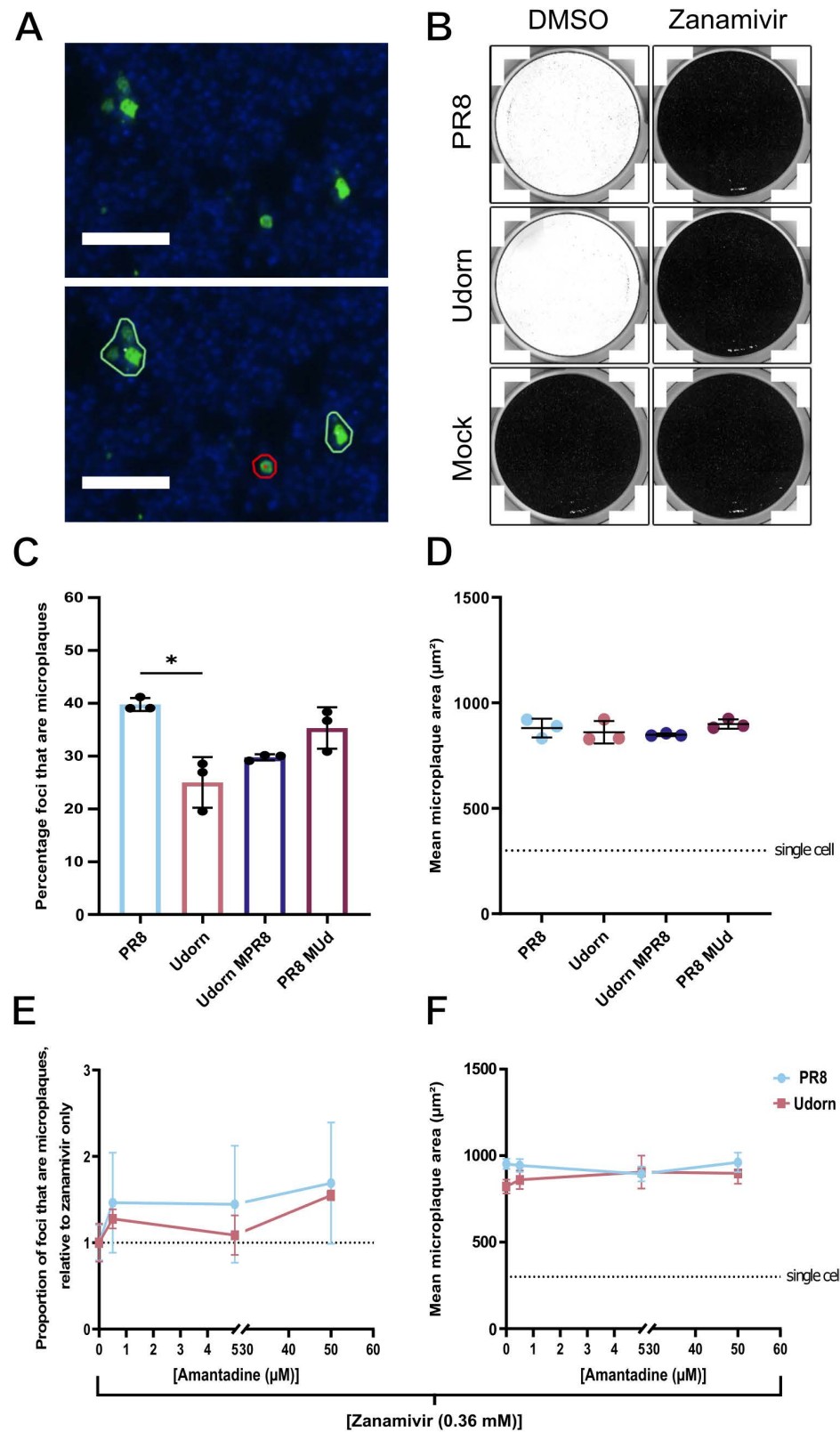

**Fig 5. The direct cell to cell spread of IAV is independent of virion morphology. (A)** Representative images of infected MDCKs at 48 hours after PR8 infection under microplaque assay conditions, imaged by a Nexcelom Celigo image cytometer. Gating thresholds were applied (bottom panel)

identifying microplaques (adjacent NP positive cells, circled in green) or isolated infected cells (circled in red). Nuclei (blue), NP (green). Scale bars = 100 μm. **(B)** Coomassie stained MDCK cells 72 h after inoculation with TPCK trypsin treated supernatants, collected from PR8 or Udorn infected microplaque assays performed in the presence of 0.36 mM zanamivir. After adsorption, the inoculum was discarded and residual zanamivir was removed by washing with PBS, after which the cells were cultured under an Avicel overlay containing 1 μg/ml TPCK trypsin. Images are representative of two biological repeats. MDCK cells were infected with WT or segment 7 reassortant viruses in the presence of 0.36 mM zanamivir for 48 hours and **(C)** the percentage of NP positive foci that are microplaques and **(D)** the mean microplaque area were determined. The significance of differences between strains was determined using a Kruskal-Wallis test (n.s. $p > 0.05$, * $p < 0.05$). MDCK cells were infected with WT PR8 or Udorn virus in the presence of 0.36 mM zanamivir and varying concentrations of amantadine for 48 hours. **(E)** The proportion of NP positive foci that are microplaques, relative to zanamivir only treatment, and **(F)** the mean microplaque area, with the dashed line indicating the mean area of a single cell. Differences between strains at each concentration were tested for significance by a Mann-Whitney test, and differences between the same strain at different concentrations were tested by Kruskal-Wallis test (n.s. $p > 0.05$). For all data the mean and SD are shown (n = 3).

mechanisms for the direct cell to cell spread of IAVs have been proposed, which could operate simultaneously: the transfer of cell associated viruses, requiring the endosomal pathway following internalisation of virions [2], and the transfer of viral genomes directly between cells with involvement of the actin cytoskeleton [1]. To distinguish between these we used amantadine, an M2 ion channel blocker which prevents virion uncoating during entry.

We first characterised the antiviral effect of amantadine in a microplaque assay (S6 Fig). We confirmed previous reports that PR8 and Udorn have some resistance to amantadine in the presence of TPCK trypsin, however, amantadine does reduce CPE in a concentration dependent fashion (S6 Fig). Once virion release is completely inhibited by zanamivir, increasing concentrations of amantadine caused no further reductions in microplaque formation or in microplaque area (Figs 5E and S5F). This suggests that the entry and uncoating of cell associated viruses (which should be inhibited by amantadine) cannot be responsible for the efficient direct cell to cell spread of PR8 and Udorn infection which we observe (Fig 5C). The alternative to this is mechanisms that can transfer cytoplasmic viral genomes directly from cell to cell, such as the transport through open-ended TLSs [5]. These results in combination with our earlier observation that PR8 infection causes a slightly greater induction of TLSs when compared to Udorn (Fig 4D), suggest that this strain variation in the frequency of direct cell to cell spread (Fig 5C) may be a result of differences in the extent of TLS induction between these viruses. However, other mechanisms of direct cell to cell spread can not be ruled out.

## IAVs induce TNT-like structures by triggering apoptosis

Thus far our data revealed that TLS induction is a common feature of IAV infection and that the triggering of this does not involve extracellular signals, is U.V. sensitive and requires replicating virus within cells. This suggested that the induction of TLSs by IAVs follows an intracellular host response to infection. One of the most striking responses of the cell to IAV infection is the onset of apoptosis, and previous reports have found that apoptosis can trigger the formation of TLSs between stressed and healthy cells [22,23]. U.V. inactivated IAVs fail to induce apoptosis [59], and as is often the case in IAV infections we observed significant evidence of apoptosis (nuclear fragmentation, membrane blebbing and apoptotic bodies) under infection conditions where TLSs were induced (Figs 3Biii, 3G and 4C). Therefore, we hypothesised that the triggering of apoptosis by IAVs is required for the induction of TLSs.

To test this, we first wanted to determine whether the triggering of TLS formation correlated with the onset of IAV induced apoptosis. To do this we performed a time course of infection with BrightFlu, a modified PR8 virus encoding the ZsGreen fluorophore [60], in MDCK cells overlaid with media containing a live cell active caspase 3/7 detection reagent. We imaged infected cells every 4 hours up to 16 hours post infection (Fig 6A) and found that, although infection (cells positive for ZsGreen) was comparably high at both 12 and 16 hours, apoptosis (detected by caspase 3/7 activation) was only observed at the latter infection timepoint (Fig 6B). These two time points (12 and 16 hours) enabled us to test the hypothesis that the onset of apoptosis coincides with TLS induction by IAVs. We had already found that TLSs were induced 16 hours post infection (Fig 4D), when infection levels and caspase activation were both high (Fig 6A and 6B). In contrast, we found that at 12 hours post infection, when infection levels were high but caspase activation was low (Fig 6A and 6B),

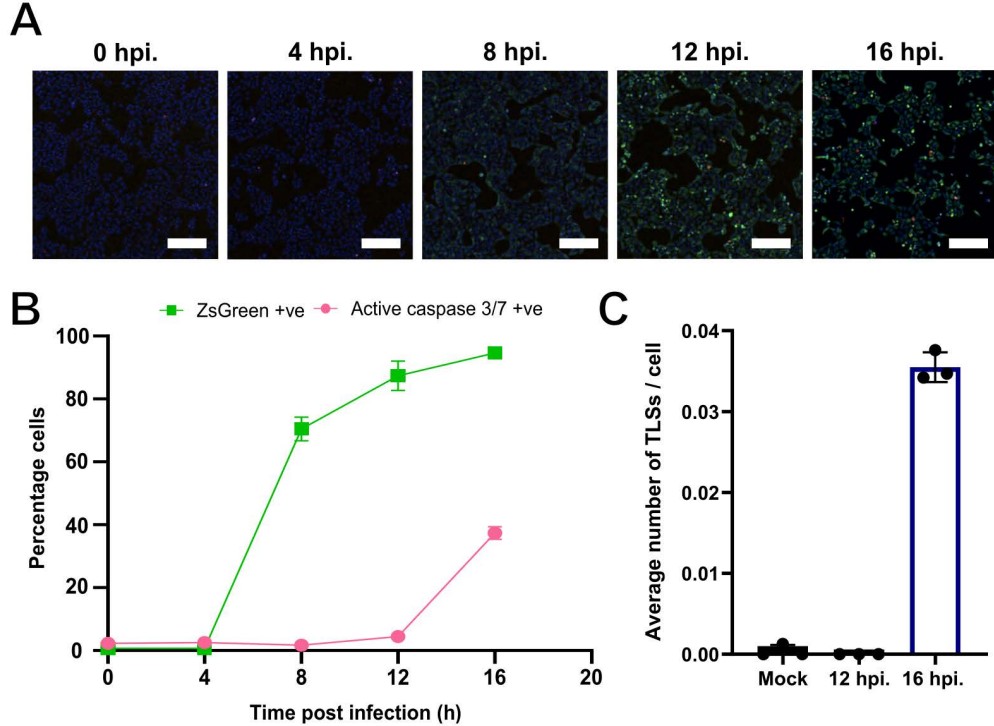

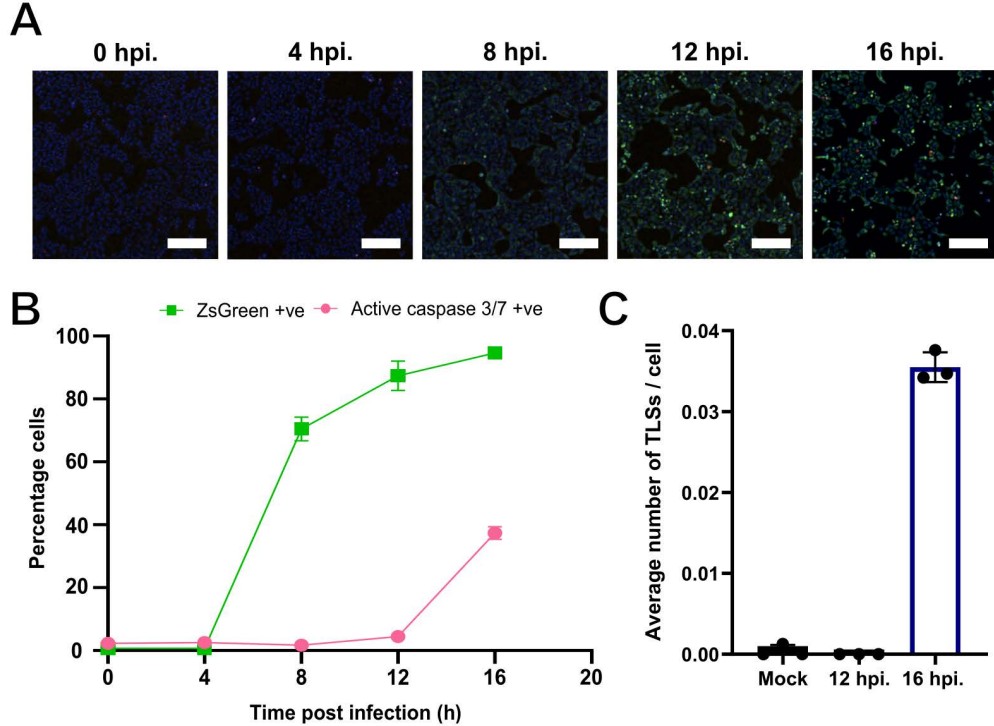

**Fig 6. TLS induction by IAVs correlates with caspase activation. (A)** Representative images of BrightFlu ZsGreen infection time course. MDCK cells were infected at an MOI of 1.5 PFU/cell, then overlaid with media containing 0.36 mM zanamivir and 60 μM active caspase 3/7 detection reagent and wells scanned live every 4 hours on the Nexcelom Celigo image cytometer. Nuclei (blue), BrightFlu ZsGreen (green), active Caspase 3/7 (red). Scale bar = 200 μm. **(B)** The percentage of cells ZsGreen positive, i.e., infected (represented by the green line), and the percentage of cells positive for active caspase 3/7, i.e., apoptotic (represented by the pink line), at each time point. **(C)** Average number of TLSs per MDCK cell 12 and 16 hours post infection with BrightFlu at an MOI of 1.5 PFU/cell. For all data the mean and SD are shown (n = 3).

there was no induction of TLSs with comparable levels to mock infection (Fig 6C). This correlation supports the hypothesis that the induction of TLSs by IAVs results from the onset of virus-induced apoptosis.

To confirm this, we sought to inhibit IAV induced apoptosis. We selected the pan-caspase inhibitor Z-VAD-fmk as it has been used previously to inhibit apoptosis of IAV infected cells [61]. We first applied Z-VAD-fmk at increasing concentrations to MDCK cells which had been infected with BrightFlu at an MOI of 1.5 PFU/cell. At 16 hours post infection, apoptotic and infected cells were imaged and classified (Fig 7A). We found that at this time point, infection resulted in approximately 95% of cells being positive for ZsGreen signal, with 35% also positive for active caspase 3/7 (Fig 7B). Treatment with increasing concentrations of Z-VAD-fmk (up to 100 μM) reduced the percentage of infected cells that were positive for active caspase 3/7 to 16% (Fig 7B). Importantly, we found that Z-VAD-fmk had no antiviral effect with the percentage of cells infected remaining consistent with the mock treated samples across all concentrations tested (~96% of cells ZsGreen positive).

We then applied 100 μM Z-VAD-fmk to BrightFlu infected (MOI 1.5 PFU/cell) subconfluent MDCKs and quantified the TLSs that formed. BrightFlu infection led to an increase in classical cytological indicators of apoptosis, such as nuclear fragmentation and apoptotic bodies (S7A Fig), and as before this correlated with an induction of TLSs (Fig 7C). However, Z-VAD-fmk resulted in a notable lack of any signs of apoptosis (S7 Fig) and the induction of TLSs also ceased (Fig 7C). Again, we found no evidence of an antiviral effect following treatment with 100 μM Z-VAD-fmk (S7A and S7B Fig). This inhibition of TLS formation in the presence of a caspase inhibitor, at a time point post infection where both apoptosis and

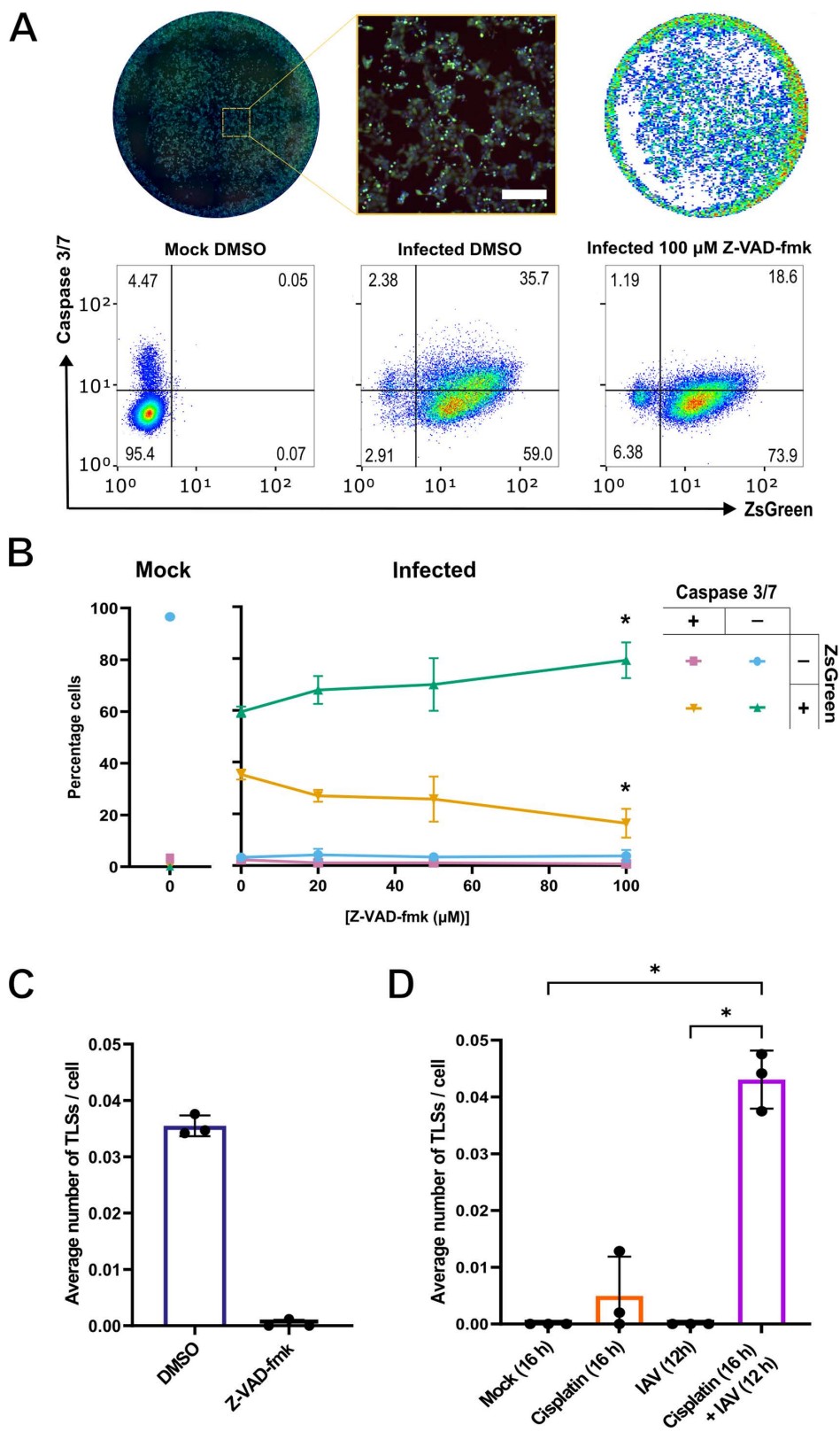

**Fig 7. TLS induction from IAV infected cells can be inhibited or achieved by the regulation of apoptosis. (A)** Representative images of an infected DMSO treated well 16 hours post infection, alongside a magnified inset (nuclei stain (blue), BrightFlu ZsGreen (green), active caspase 3/7 (red).

Scale bar = 200 μm) and the nuclei masks applied during the analysis pipeline following imaging (upper panel). Representative FlowJo analyses plots (lower panel) of mock or PR8-ZsGreen (BrightFlu) infected MDCK cells, treated 1 hour post infection with either DMSO (mock) or 100 μM Z-VAD-fmk. Cell populations were classified according to the expression of ZsGreen as a marker of infection, and the presence of active caspase 3/7 as a marker of apoptosis. Gating's were established according to a mock DMSO control in the absence of active caspase 3/7 detection reagent. **(B)** The percentage of MDCK cells positive or negative for active caspase 3/7 and ZsGreen, at 16 hours post infection with BrightFlu at an MOI of 1.5 PFU/cell, in the presence of 0.36 mM zanamivir and DMSO (mock) or increasing concentrations of Z-VAD-fmk (added 1 hour post infection). The percentage cells negative for both red and green signal is shown by the blue line, uniquely red or green by the pink and green lines respectively, and the yellow line represents the percentage cells positive for both red and green signal. The reductions in the percentage of cells singly positive for ZsGreen, or doubly positive for caspase 3/7 and ZsGreen, when compared to the infected DMSO control, were tested for significance using a Kruskal-Wallis test (n.s. $p > 0.05$, *$p < 0.05$). **(C)** Average number of TLSs per MDCK cell at 16 hours post infection with BrightFlu at an MOI of 1.5 PFU/cell in the presence of DMSO or 100 μM Z-VAD-fmk (added 1 hour post infection). **(D)** Average number of TLSs per MDCK cell 16 hours post treatment with 30 μM cisplatin in the presence or absence of an IAV infection (MOI 1.5 PFU/cell) performed for 12 hours. Data relating to the single 12 hour infection is replotted from Fig 6. Differences in the average number of TLSs/ cell between conditions was tested for significance by Kruskal-Wallis test (n.s. $p > 0.05$, *$p < 0.05$). For all data the mean and SD are shown (n = 3).

TLS induction is otherwise seen, strengthened our conclusions that the induction of TLSs by IAVs requires the triggering of apoptosis.

To investigate whether apoptosis alone was responsible for the induction of TLSs by IAVs, we tested the effect of cisplatin. Cisplatin is a chemotherapeutic drug known to induce apoptosis within cancer cells (including MDCK cells) through the activation of caspases [62]. We found that treatment with 30 μM cisplatin significantly induced caspase 3/7 activation in cells at 16 hours post treatment (S8A Fig). We then utilised these conditions to study TLS induction from uninfected, cisplatin treated, subconfluent MDCK cells. We found that cisplatin treatment did not significantly induce the formation of TLSs above mock treated levels (Fig 7D), despite notable cytological signs of apoptosis (S8B Fig). We hypothesised that although apoptosis is required for IAVs to induce TLS formation, apoptosis alone is not sufficient. This is consistent with our data showing that TLS formation requires both replication competent virus, and the activation of apoptosis during infection (Figs 3C and 6).

Based on this hypothesis, we wondered if cisplatin treatment could increase TLS formation at an earlier infection time point. To explore this, we treated MDCK cells with 30 μM cisplatin for 4 hours, then infected the same cells with an IAV before adding the cisplatin-containing overlay for a further 12 h. Individually, cisplatin treatment and earlier IAV infection did not lead to an induction of TLSs (Fig 7D). However, when both conditions are applied in combination, TLSs are induced (Fig 7D). This therefore confirms that TLS induction by IAVs requires both intracellular virus replication and the triggering of apoptosis.

### Inhibition of apoptosis reduces the ability of IAVs to directly spread to distant cells

Our results demonstrate that Z-VAD-fmk is a potent inhibitor of TLSs (Fig 7C), and we hypothesised that this drug would therefore inhibit the direct cell to cell spread of IAV infection. To investigate this, we first used a microplaque assay (Fig 5), noting that as this involves confluent cells, multiple forms of direct cell to cell spread are in principle possible in this assay. We performed the microplaque assay as done prior, but this time included different TLS inhibiting drugs within the overlay added to cells post virus adsorption. This included the apoptosis inhibitor Z-VAD-fmk; cytochalasin D and IPA-3, which were selected as they have both been reported to inhibit TNT formation via causing F-actin depolymerisation [1]; and taxol (Paclitaxel) as its microtubule stabilising effect has been shown to inhibit TLSs [1], most likely through inhibiting the formation of a subset of TNTs that also incorporate microtubules [11,63]. We confirmed that the concentrations of drugs did not interfere with the cytoplasmic localisation of NP at 16 hours post infection, suggesting that the virus replication cycle and vRNP trafficking was not disrupted (S9A Fig). Furthermore, cells treated with TLS inhibiting drugs displayed similar cytological effects to those previously reported (S9A Fig) [1]. Due to the severe alteration of the cytoskeleton following cytochalasin D and taxol treatment, we were only able to reliably score TLSs from infected MDCK cells treated with IPA-3, which showed a significant reduction in TLS formation when compared to the infected DMSO control (S9B Fig).

In line with our previous findings (Figs 5 and S5) the inclusion of zanamivir in the absence of TPCK trypsin conferred a consistent, though non-significant reduction in the frequency and scale of direct cell to cell spread (Fig 8). This reduction was seen across drug treatments, with the exception of cytochalasin D and taxol. Intriguingly, these drugs also increased the frequency of direct cell to cell spread of infection, with the increase of the former being statistically significant (Fig 8A). An increase was also seen in the mean area of microplaques following cytochalasin D treatment, even though when comparing mean microplaque areas between drugs in combination with zanamivir there were no differences (Fig 8B). This suggests that under the conditions of this assay, cytochalasin D and taxol increases the ability of single infected cells to spread infection to neighbouring cells. The fact that these drugs, each with a different mechanism of action, do not reduce the capacity of the virus for direct cell to cell spread suggests a number of scenarios. First, TLSs may not be able to form between confluent cells *in vitro* and the presence of TLSs connecting cells in a confluent monolayer can not be verified. Secondly, this assay is likely conducive to several mechanisms of direct cell to cell spread that can function between cells in close contact and are not reliant on distant cell contacts such as TLSs. It is unknown how these drugs are negatively or positively influencing these alternative mechanisms of direct cell to cell spread.

Given that the microplaque assay cannot attribute direct cell to cell spread specifically to TLSs, we implemented an alternative assay using the subconfluent MDCK cell culture conditions we had previously shown to be conducive to TLS formation (Figs 4D and 6C). By using subconfluent cells we limited the mechanisms of direct cell to cell spread to only those that can deliver viral genomes to distant cells, a feature known to be conferred by TLSs [64]. To study the effect of TLS inhibiting drugs on the direct cell to cell spread of infection between subconfluent cells, we prevented virion mediated spread, as done previously (Fig 5), and determined the percentage of cells infected with BrightFlu ZsGreen above infection input levels by flow cytometry, and across drug treatments (Fig 8C). Amongst the drugs targeting F-actin, we found that cytochalasin D had no effect on the percentage of cells infected, whereas IPA-3 caused a moderate but consistent reduction (Fig 8C). The microtubule targeting drug taxol resulted in an even greater reduction in the percentage of cells infected when compared with IPA-3 (Fig 8C). This reveals that these drugs differ in their inhibitory effect on direct cell to cell spread when administered at concentrations that are neither directly antiviral or cytotoxic. TLS targeting drugs disrupting both F-actin (IPA-3) and microtubules (Taxol) can inhibit direct cell to cell spread of IAV infection (Fig 8C) [1]. However, the greatest reduction in the percentage of cells infected was seen with the anti-apoptotic drug Z-VAD-fmk (Fig 8C), with infection largely being unable to directly spread to any neighbouring cell as indicated by only a slight increase above infection input levels. This result correlated with our observation that Z-VAD-fmk resulted in an even greater reduction in TLS formation than IPA-3 (Figs 7C and S9B). These results, when combined with our findings on the role of apoptosis in triggering TLS formation during IAV infection, provide strong evidence that IAVs use this cell death response to facilitate the direct cell to cell spread of infection between non-adjacent cells via long-range cell connections.

## Discussion

The formation of TLSs has been shown to mediate the direct cell to cell spread of IAV infections *in vitro* [5]. However, it was unknown if this effect could occur in the crowded, mobile environment of the respiratory tract, and in particular the dynamic pseudostratified respiratory epithelium, or how IAV infection induced the formation of TLSs.

Here, we provided the first evidence that TLSs do form at the sites of IAV infections, using an *in vivo* reporter mouse system to demonstrate that TLSs can form from IAV infected cells in the airway epithelium. This shows that TLSs have the potential to contribute to the within-host spread of IAV infections and also expands the range of tissues known to be conducive to TLS formation [25].

Our data shows no evidence that low MOI IAV infections establish the conditions for TLS pathfinding (Fig 2), and we show that in a co-culture, uninfected cells produce TLSs at a similar frequency to IAV infected cells (Fig 2B), in line with previous studies [3,38,65]. Because of this, the contribution of TLSs to IAV infection spread or coinfection is likely driven by the local density of infected cells at any given point in time.

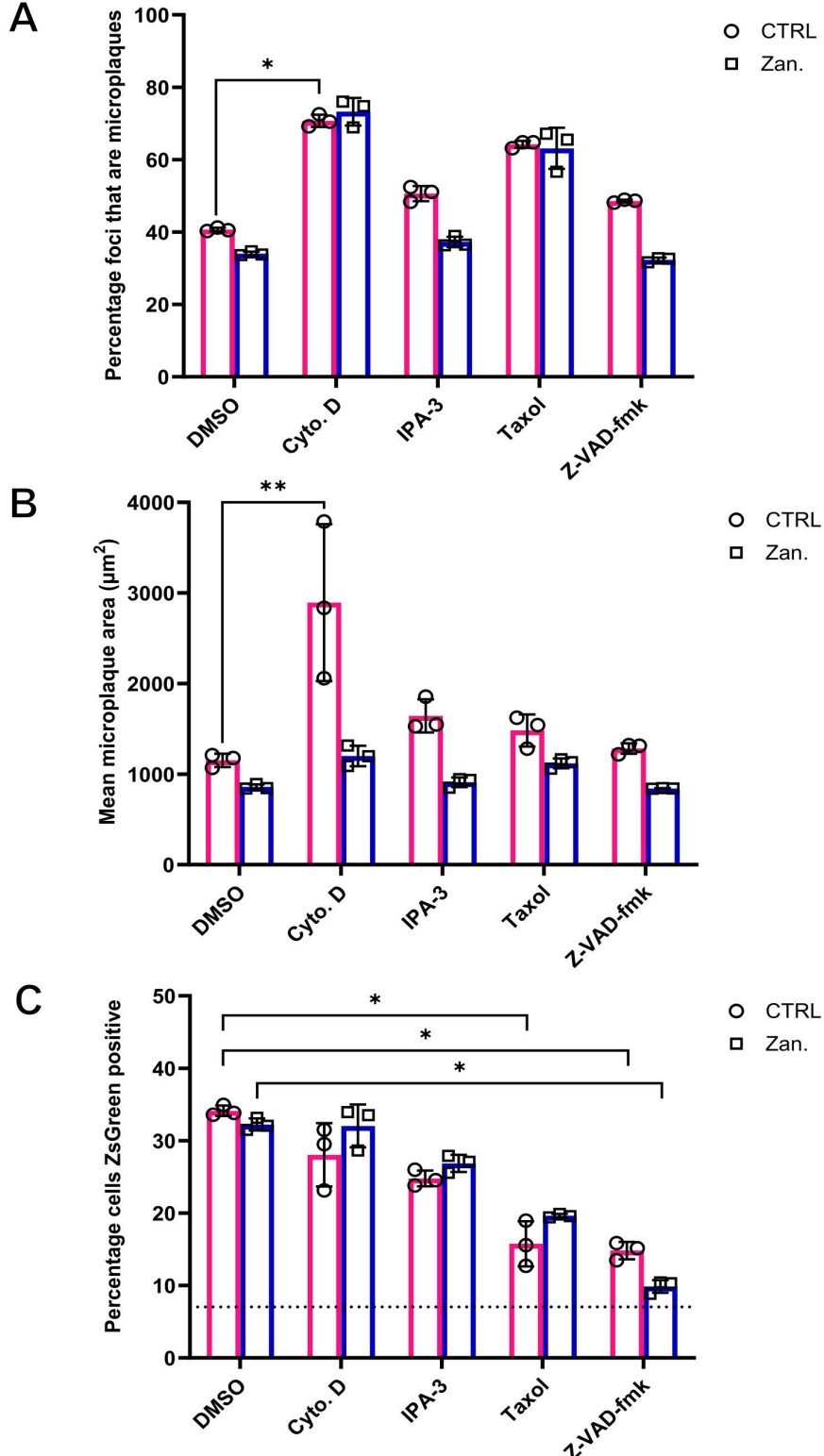

**Fig 8. Z-VAD-fmk inhibits the direct cell to cell spread of IAV infection between subconfluent cells. (A)** The percentage of NP positive foci that are microplaques in the absence of TPCK trypsin and **(B)** the mean area of microplaques with and without 0.36 mM zanamivir treatment, in the presence of a panel of TLS inhibiting drugs. Confluent MDCKs were infected with PR8 at a low MOI, and 2 hours post infection an overlay containing

either DMSO, cytochalasin D (20 µM), IPA-3 (3 µM), taxol (100 µM), or Z-VAD-fmk (100 µM) was added. Cells were fixed 48 hours post infection, and imaged on the Nexcelom Celigo image cytometer. **(C)** The percentage of cells positive for ZsGreen, as analysed by flow cytometry, 24 hours post PR8-ZsGreen (BrightFlu) infection (MOI 0.1 FFU/cell) of subconfluent MDCK cells in the presence or absence of 0.36 mM zanamivir and a panel of TLS inhibiting drugs. The dashed line indicates the percentage cells ZsGreen positive 8 hours post infection, i.e., infection levels prior to any infection spread. Differences between mock and zanamivir treatments was tested for significance using a multiple Mann-Whitney test (n.s. $p > 0.05$). Differences between TLS inhibiting drug treatments relative to the DMSO control, either as part of the control or zanamivir treated group, was tested for significance using a Kruskal-Wallis test (n.s. $p > 0.05$, $*p < 0.05$, $**p < 0.01$). For all data the mean and SD is shown (n = 3).

We then used *in vitro* assays to identify the factors behind TLS induction following high MOI infections. We found that the induction of TLSs is driven by the presence of replicating virus. This effect was not mediated through paracrine signalling between cells or through IAV's ability to form filamentous virions that resemble TLSs. Instead, IAVs induce TLS formation through intracellular virus replication, triggering apoptosis (Figs 6 and 7). Apoptosis was already known to contribute to the pathology of IAV infection [66], and our study shows that it can also be used by the virus to facilitate its direct cell to cell spread.

Apoptosis can be distinguished from other cell-death programmes such as necroptosis and pyroptosis by its reliance on caspases, and by the limited involvement of extracellular pathogen- or damage associated molecular patterns, respectively, as stimuli (reviewed in [67,68]). Recent studies have demonstrated that caspase 8 is also involved in the activation of pyroptosis, in a process independent of extracellular PAMPS or TNF (reviewed in [68]). However, there are redundant pathways for pyroptosis activation, with nuclear programmed-death ligand 1 (nPD-L1) also capable of progressing the cascade [69]. Furthermore, treatment of IAV infected dendritic cells and mouse embryonic fibroblasts with a pan-caspase inhibitor did not prevent necroptosis [70,71]. Therefore, the pan-caspase inhibitor Z-VAD-fmk can be considered primarily as an inhibitor of apoptosis in our experiments, in line with its use in other studies [22,61].

How apoptosis induces TLS formation is complex, as multiple stages of apoptosis can be important for TLS induction. For example, one study found that mitochondrial cytochrome c release within U.V. treated PC12 cells induced the formation of microtubule-containing TLSs [22]. Another study found that the presentation of phosphatidylserine (PS) on the outer leaflet of the plasma membrane was required [23]. Coating PS with annexin V prevented TLS induction from stressed cells but had no effect on the frequency of other cellular projections [23]. This suggests that the presentation of PS on the extracellular surface could increase the rate at which cell projections (or TLS precursors) can interact with recipient cell membranes, thereby completing and inducing the formation of TLSs. This could also extend to the observation that uninfected cells were capable of initiating TLS connections under low MOI conditions (Fig 2G). We hypothesise that under such conditions the presentation of PS from infected and apoptotic cells could mediate more intercellular membrane interactions with cell projections originating from uninfected cells. This is supported by the finding that there is a greater likelihood for TLS connections to involve at least one infected cell (approximately 60% of TLS connections involved at least one infected cell despite infection of only 37% of cells, S3 Fig).

Apoptosis is a cell death programme which is already known to positively influence IAV virus titres *in* vitro and can be seen *in vivo* following IAV infection [59,72,73]. IAVs have been shown to modulate apoptosis in ways that can benefit virus propagation [59,74]. The increased formation of TLSs following the onset of apoptosis could also benefit the virus by facilitating the direct cell to cell spread of IAV genomes, and so spreading IAV to a healthy new host cell that can continue viral replication. There may also be other advantages to the virus – for example, it has been shown that mitochondrial trafficking through TLSs can rescue porcine reproductive and respiratory syndrome virus (PRRSV) infected cells from death by apoptosis [22,23,75]. It would be interesting to see if IAV infection also promotes intercellular exchange of mitochondria through TLSs, and if this could delay cell death and prolong IAV replication within the primary infected cell.

However, the lack of TLS induction by cisplatin treatment in the absence of an IAV infection suggests that the triggering of apoptosis alone is not sufficient for TLS induction. Cisplatin induces apoptosis in a broadly similar way to IAV infection, as both can trigger the intrinsic and extrinsic apoptotic pathways (reviewed in [76,77]). Unique to the mechanism

of apoptosis by IAV infection is the function of viral proteins (reviewed in [76]), most notably PB1-F2, which induces the permeabilisation of mitochondrial membranes leading to cytochrome c release and the activation of the intrinsic pathway [78,79]. Our results showed that when apoptosis is triggered pharmacologically, TLSs can be induced from infected cells at earlier time points post infection (Figs 6 and 7D). These results collectively reveal that the mechanism of TLS induction by IAVs is due to a combination of virus replication and the onset of apoptosis, and therefore, potentiates an involvement of IAV proteins that either directly or indirectly influence this cell death response.

For some other viruses, viral proteins are known to be directly involved in the induction of TLSs (reviewed in [11]). For example, the Nef and Env protein of HIV-1, as well as the US3 proteins of the alphaherpesviruses, pseudorabies virus (PRV), Herpes-Simplex virus 2 (HSV-2), and bovine herpesvirus 5 (BoHV-5), have been shown to trigger TLS formation [63,65,80–82]. We note that these proteins are also known to regulate apoptosis [63,82–85]. Interestingly, in the case of HIV-1 Nef and US3 of alphaherpesviruses, these proteins have the ability to negatively regulate apoptosis [82,83,86,87].

Here, we show that drugs that inhibit TLS formation reduce the ability of IAVs to spread directly between subconfluent cells. This strongly supports a functional role of apoptosis in facilitating the transfer of IAV infection to neighbouring, distant cells via TLSs (Fig 8C). Whilst previous reports showed that several of these drugs reduced direct cell to cell spread of IAV infection between confluent cells [1,5], in our hands these drugs did not reduce the frequency and scale of microplaque formation (Fig 8A and 8B). Nevertheless, we confirm previous reports that direct cell to cell spread is a function that can be performed by many IAV infected cells (up to 40%, Fig 5) [1]. This efficient direct cell to cell spread is also consistent with reports of other respiratory viruses, including HMPV (40% of HMPV infected BEAS-2B cells directly spread infection, 24 hours post coculture) [88], and RSV (approximately 15% of RSV infected HEp-2 cells, 24 hours post infection) [89].

We show for the first time that the efficiency of direct cell to cell transfer of cytoplasmic viral genomes varies between IAV strains (PR8 and Udorn, Fig 5). This was not influenced by the differences in virion morphologies of the viruses tested, suggesting that other factors are responsible (Fig 5C and 5D). We note that the differences in direct cell to cell spread between viruses correlated with the extent by which they induced TLSs. However, other mechanisms of direct viral genome transfer can not be ruled out. It is possible that these viruses interact differently with host factors that traffic viral genomes. For example, in the case of TLS mediated IAV direct cell to cell spread, Rab11a is required for the incorporation and trafficking of NP (used as a proxy for vRNPs) [5]. Although, the lack of a difference in the amount of NP positive TLSs (Fig 4E) suggests that this particular factor may not account for the differences observed here [5].

In this study, we examine how IAVs establish direct cell to cell spread, allowing them to seed new infections even when virion spread is inhibited. We report for the first time structures resembling TNTs within the lungs of infected mice, suggesting that TLS-mediated spread of infections could occur within the respiratory epithelium. We also show that replicating IAVs induce TLS formation by driving cells into apoptosis, and that this host response was required for the direct cell to cell spread of IAV infection between distant cells. These data suggest that virus-induced apoptosis plays a previously unappreciated role in the spread of IAV, and potentially in the spread of many other viruses, within the tissues of their infected hosts.

## Materials and methods

### Ethics statement

All animal work was done in strict accordance with the EU Directive 2010/63/eu and Animal (Scientific Procedures) Act 1986. Procedures were approved by the Cancer Research UK Scotland Institute under a project licence P72BA642F.

### Cells, plasmids and viruses

Madin-Darby Canine Kidney (MDCK) cells, human embryonic kidney 293T cells (HEK293T), and adenocarcinoma 549 (A549) cells were maintained in Dulbecco's Modified Eagle Medium (DMEM, Gibco) supplemented with 10% Foetal Bovine Serum (FBS, Gibco). Cells were maintained at 37°C and 5% $CO_2$ in a humidified incubator.

A549 AcGFP1 cells were generated using prepackaged lentivirus (TakaraBio, rLV.EF1.AcGFP1-Mem-9). Transduction of A549 cells was performed as per the manufacturer's instructions. Briefly, A549s were seeded into a 6 well plate at a density of $2.5 \times 10^5$ cells per well. Transduction mix was prepared by adding lentivirus to complete media containing polybrene (4 µg/ml, Sigma) to achieve an MOI of 10. The transduction mix was added directly over cells and incubated for 5.5 h at 37°C. The cells were washed once with PBS and complete media added. Following a 48 hour incubation the cells were treated and maintained with complete media supplemented with 2 µg/ml puromycin (Thermo Fisher) selecting for transduced cells.

Wild-type A/Puerto Rico/8/1934 H1N1 (PR8), and A/Udorn/307/1972 H3N2 (Udorn) P0 viruses were generated as previously described [90]. Briefly, HEK293T cells were transfected with the 8 plasmid pDUAL (kindly provided by Prof. Ron Fouchier, Erasmus MC) and 12 plasmid pHH21 (kindly provided by Prof. Paul Digard, Roslin) reverse genetic systems, respectively, and propagated on MDCK cells. Segment 7 reassortant viruses containing the matrix gene of PR8 or Udorn within a background of Udorn (Udorn MPR8) or PR8 (PR8 MUd), were prepared by swapping the corresponding vRNA encoding plasmids. BrightFlu (PR8 virus encoding the fluorophore ZsGreen in segment 8 of the genome, [60]), and PR8-Cre (PR8 virus encoding the Cre recombinase, kindly provided by Professor Ben tenOever) stocks were propagated on MDCK cells. The propagation of all viruses was performed in viral growth media (VGM) (serum-free DMEM supplemented with 1 µg/ml TPCK-treated Trypsin and 0.14% Bovine Serum Albumin (BSA) Fraction V, Sigma) except where stated otherwise. For viral stocks, media were collected 48 hours post infection and cell debris removed by centrifugation at 3000 rpm for 5 minutes; aliquots were stored at -70°C. Infectious titres of viruses (as plaque forming units (PFU)/mL) were determined under agarose by conventional plaque assay in MDCKs [91].

The infectious titre of BrightFlu virus in fluorescent forming units (FFU) per mL, was also determined by serially diluting virus in VGM and absorbing onto subconfluent MDCK cells for 45 minutes. After this the inoculum was removed, and cells were washed twice in PBS and then overlayed with complete media. At 16 hours post infection, the cells were resuspended into a single-cell suspension in TrypLE trypsin (Gibco), and fixed with 4% formaldehyde. The percentage of cells positive for ZsGreen signal was determined by flow cytometry (see below) and the infectious titre calculated assuming that infections occurred independently, such that the ratio of FFU to cells could be calculated from the proportion of infected cells using the Poisson distribution [92].

## IAV infection of subconfluent cells

MDCK cells were seeded onto 13 mm glass coverslips at a density of $6 \times 10^4$ cells. Following overnight incubation, cells were washed with PBS and infected with WT or reassortant virus at an MOI of 1.5 PFU/cell. Adsorption was performed for 45 minutes AT 37°C in VGM, after which the inoculum was removed, and cells washed with PBS to remove any remaining TPCK trypsin. The cells were overlaid with 1 ml Opti-MEM (Gibco) and sixteen hours post infection formaldehyde was added directly to the Opti-MEM overlay to a final concentration of 4%. Fixation was performed at 8°C for 2 h, and coverslips were then allowed to air dry for approximately 10 minutes before washing once with 2% (v/v) FBS/PBS.

For the A549 and A549 AcGFP1 coculture, cells were seeded at a 1:1 ratio to a density of $4.5 \times 10^4$ cells. Following overnight incubation, cells were infected with PR8 at an MOI of 2.5 PFU/cell with the overlay conditions and fixation method the same as previously stated.

For the measurement of direct cell to cell spread between subconfluent cells, 24 well plates were seeded with MDCK cells as detailed above. Following overnight incubation, cells were washed with PBS and infected with BrightFlu virus at an MOI of 0.1 FFU/cell. Following a 1-hour adsorption, the inoculum was removed and cells washed twice with PBS. An overlay consisting of serum free media, with and without 0.36 mM Zanamivir was added over cells. Where indicated DMSO, or TLS inhibiting drug (cytochalasin D (20 µM, Abcam), IPA-3 (3 µM, Sigma), taxol (100 µM, Merck), or Z-VAD-fmk (100 µM, Promega)) was included in the cell overlay. At 24 hours post infection the cells were washed with PBS, resuspended into a single-cell suspension with TrypLE trypsin (Gibco), fixed in 4% formaldehyde and the percentage of cells infected (ZsGreen positive) determined by flow cytometry (see below).

## Mice infections and thick tissue sectioning

Three mT/mG mice, between 17 and 18 weeks old, were kindly provided by Dr Stephanie May (Cancer Research UK Scotland Institute). Two mice were intranasally infected with 1000 PFU of PR8-Cre. A naïve mouse was mock infected with PBS in a similar manner. Six days post infection the mice were euthanised, the lungs harvested and inflated with agarose. Lungs were fixed in 4% formaldehyde overnight before being transferred to PBA (1% BSA, 0.05% NaN$_3$ in PBS). Dissected lung lobes were embedded in optimal cutting temperature compound (OCT, VWR) with 100 µm tissue sections cut at -20°C using a CryoStat (CM1950, Leica) and slides stored at -70°C. Prior to imaging, samples were thawed, and OCT removed by immersing in PBS. Coverslips were placed on seal-frame incubation chambers (Thermo Fisher) containing Ce3D tissue clearing solution.

## Immunocytochemistry and immunostaining of virions

For virion staining, MDCKs were infected with P1 virus at an MOI of 0.25 PFU/cell. 48 hours post infection the supernatant was collected, and cell debris removed by centrifugation at 13,000 rpm for 1 minute. The virus supernatant was diluted 1:10 in PBS and spun onto coverslips by centrifugation at 1000 $g$ for 30 minutes at 4°C. Coverslips were fixed for 10 minutes with 4% formaldehyde and blocked in 2% FBS/PBS. For all viruses, with the exception of PR8 MUd, coverslips were stained with mouse anti-HA primary antibody [1:2000] (kindly provided by Dr Stephen Wharton, Francis Crick Institute) and goat anti-mouse Alexa-Fluor 555 (Thermo Fisher). PR8 MUd was stained using rabbit anti-H1 (kindly provided by Prof. Paul Digard, University of Edinburgh [93]) [1:100]. For HA surface-stained cells, unpermeabilised cells were blocked with 2% FBS/PBS for one hour with immunostaining performed as above with the inclusion of 4′,6-diamidino-2-phenylindole (DAPI, Thermo Fisher) [1:1000].

To achieve intracellular staining, cells were permeabilised with 0.2% Triton-X100 in PBS for 7 minutes and washed three times with 2% FBS/PBS. Samples were blocked in 2% FBS/PBS for 1 h followed by the addition of a sheep anti-NP antibody (PR8 H1N1 available from the Influenza Virus Toolkit at www.influenza.bio) [1:1000] and incubated for 1 h at room temperature. Cells were washed three times before secondary antibody incubation for 30 minutes using anti-sheep alexa fluor 488, 555 or 647 (Thermo Fisher) alongside DAPI [1:1000]. For super resolution confocal microscopy, cells were additionally stained with 1X phalloidin-iFluor 488, 555 or 647 (abcam) in 1% bovine serum albumin (BSA) for 30 minutes. Coverslips were mounted onto slides using ProLong Gold Antifade mounting media.

## Microplaque assay

Confluent MDCK cells were infected with PR8, Udorn or Udorn MPR8 at MOIs achieving approximately 1000 nucleoprotein (NP) positive infected foci per well. Following a 2 h incubation, the inoculum was removed, and cells acid washed (PBS-HCl pH 3) for 1 minute to inactivate free virions. Cells were washed three times with PBS before adding 1 ml overlay containing 1.2% Avicel, DMEM, 0.14% BSA and between 0.12 to 0.48 mM zanamivir (Sigma). Where indicated, amantadine (1-adamantanamine, Thermo Fisher) was included at concentrations ranging from 0.5 mM to 50 mM in place, or in combination with 0.36 mM Zanamivir. Cytochalasin D (20 µM, abcam), IPA-3 (3 µM, Sigma), taxol (100 µM, Merck), or Z-VAD-fmk (100 µM, Promega) was also included in the overlay where indicated.

At forty-eight hours post infection, the overlay was removed by washing with PBS. Cells were fixed with either 4% formaldehyde for 10 minutes or Coomassie Blue for 30 minutes.

## U.V. inactivation of virus and western blotting

Supernatants were collected and distributed in 100 µl volumes in a 96-well plate. Inactivation of virus was performed using an 8W 254 nm U.V. lamp (UVS-28 EL, UVP) placed directly above samples whilst on ice for 6 minutes, with shaking every 2 minutes. U.V. treated samples were collected and added directly to fresh cells.

At specified time points cells were harvested and lysed in Laemmli buffer (20% glycerol, 2% SDS, 24 mM Tris pH 6.8, 0.1M DTT, 0.2% bromophenol blue, 0.2% cyanol) supplemented with benzonase (Merck) and protease inhibitor cocktail (Thermo Fisher). Samples were heated at 37°C for 30 minutes, then at 95°C for 5 minutes. Samples were run on AnyKD mini-PROTEAN TGX gels (BioRad) before being transferred to a nitrocellulose membrane using the TransBlot Turbo quick transfer protocol. Membranes were blocked for 1 h with 0.1% Tween 20 (PBS-T)/5% milk, washed three times with PBS-T and stained overnight at 4°C in PBS-T/5% milk containing primary antibodies (anti-phosphorylated STAT1, Cell Signalling Technology and anti-alpha Tubulin, Merck) [1:1000]. The membranes were washed three times with PBS-T before secondary antibody incubation with anti-rabbit DyLight 800 and anti-Mouse DyLight 680 (Invitrogen) diluted in PBS-T/5% milk [1:10,000] for 45 minutes. Membranes were washed three times with PBS-T, PBS and water prior to imaging using the LI-COR CLx-Odyssey Imaging platform. Quantification was performed by measuring band intensities in Image Studio Lite Software, with the α-tubulin loading control used for normalisation.

### Detection of active caspase 3/7 and regulating apoptosis

MDCKs were infected at an MOI of 1.5 PFU/cell with BrightFlu. Following virus adsorption, an overlay consisting of DMEM, 60 µM CellEvent caspase 3/7 red detection reagent (Thermo Fisher), and DMSO was added. Where indicated, the pan-caspase inhibitor Z-VAD-fmk (Promega) was also included in this overlay to the stated final concentrations. Cells were incubated for the specified time points. Hoescht 33342 was added directly to the overlay achieving a final concentration of 5 µg/mL and incubated for 20 minutes prior to imaging. Cells were imaged live using the Nexcelom Celigo image cytometer.

For the induction of apoptosis, cisplatin (Sigma) was diluted to the specified concentration in complete media. When caspase activity was to be measured, 60 µM caspase 3/7 red detection reagent (Thermo Fisher) was also included to the overlay added to subconfluent MDCK cells. Sixteen hours post treatment, cells were either stained with Hoechst 33342 and imaged live as done prior, or fixed for 2 hours at 8°C in 4% formaldehyde as described above.

### Confocal microscopy

Confocal microscopy was performed using the Zeiss LSM 880 (63x oil immersion objective, 1.4 NA). Super resolution imaging of TLSs and budding filaments was performed using Airyscan fast detection. Post-acquisition auto processing was performed within Zen Black (Zeiss) software (v14.0.29.201). For TLS quantification, a single field of view encompassed two adjacent tiles stitched together. In total 14 randomly selected fields of view with a suitable distribution of cell nuclei were selected per technical replicate. Each biological replicate consisted of two technical replicates. Imaging of filaments on coverslips was performed similarly except for the sole use of the Gallium Arsenide Phosphide (GaAsP) detector.

Thick section confocal microscopy was similarly performed utilising Z-stacks encompassing 3D regions of interest. Maximum intensity projections of 3D airyscan processed Z-stacked images were created using the Zen Black (Zeiss) software (v14.0.29.201).

### Image analysis

Three-dimensional surface renders of fluorescent objects within thick tissue sections were created from Z-stack images via the creation of binary masks on Imaris (Andor). Any further background removal required was performed within Imaris according to samples derived from naïve mice.

Filament measuring was performed using custom image J macro scripts published previously [94]. Briefly, micrographs were auto thresholded, and debris with a circularity between 0.5 and 1 was removed via Particle Remover. The dimensions of quantified remaining particles were extracted with Ridge Detection and figures created on GraphPad prism.

TNT-like structures (TLS) were quantified manually using characteristic features of TNTs that differentiate them from other protrusions. These include the presence of a narrow structure that appears to connect two or more cells and

contains F-actin along its length. This excludes the false classification of nanopodia which typically lack F-actin [95]. Structures also had to exceed a minimal length threshold of 5 µm to be positively classified as a TLS to help distinguish them from filopodia that rarely extend beyond this length [64]. The structure must connect cells which have a visibly intact nucleus and are not showing signs of recent cell division, e.g., cellular midbodies, as have been done in previous analysis [38,96]. Quantification of micrographs was performed blind to the experimental condition where possible to avoid analyst bias.

Microplaque image analysis was performed on the Nexcelom Celigo image cytometer using a 90% well mask. A gating area of 600 µm$^2$ was selected on the Celigo analysis software to distinguish microplaques (i.e., areas with Alexa fluor 488 signal > 600 µm$^2$) from single infected foci. This area was optimised on the Celigo by manually adjusting the gating size till single infected cells (as determined by both NP and DAPI staining) were classified uniquely from fluorescent regions encompassing at least two adjacent infected cells. The percentage of total foci that existed as microplaques was determined alongside the mean microplaque area as measured by the Celigo.

The detection of cells positive for active caspase 3/7 and ZsGreen was performed on a Celigo imaging cytometer. The Hoechst stain was used to apply a nuclei mask within the analysis software on the Celigo imaging cytometer. A dilation radius of 5 µm was applied to capture perinuclear and cytoplasmic red or green signal. The percentage of cells negative, singly positive, or dual positive for active caspase 3/7 and ZsGreen was performed within FlowJo software (v10.10). Cell population gating was established based on mock infected nuclei-stained controls and then applied to all samples.

### Flow cytometry

Cells positive for ZsGreen signal were detected and quantified using a Guava easyCyte HT cytometer (Luminex). FlowJo software v10.10 was used to analyse the data by first isolating single cells and adjusting the gating of mean fluorescence intensities according to a mock infected sample. This then established the mean fluorescence intensity thresholds used to classify cells as positive or negative for ZsGreen expression.

### Supporting information

**S1 Fig. A TNT-like structure connecting IAV infected epithelial cells within the lungs of a reporter mouse.** Maximum intensity projection of a TNT-like structure (TLS) within thick sectioned lung tissue connecting IAV infected cells. Magnified inset outlined in a white box is shown alongside a 3D render (bottom right panel). Cell membranes are labelled in tdTomato (magenta) for uninfected cells, and GFP (green) for infected cells. White arrow indicates the presence of a TLS. Scale bars = 20 µm.
(TIFF)

**S2 Fig. A549 AcGFP cells demonstrate similar properties to WT A549 in infected coculture.** (A) Enlarged split channel images of A549 AcGFP1 cells cocultured with WT A549 cells at a ratio of 1:1 and infected at a low MOI. 16 hours post infection the cells were fixed and immunostained for NP. Nuclei (yellow), F-actin (magenta), AcGFP (green), NP (blue). Scale bar = 50 µm. (B) Following manual assessment of micrographs, the percentage of cells in coculture that were GFP positive (i.e., A549 AcGFP), GFP positive and infected and the percentage of TLS consisting of GFP labelled membrane throughout its length was determined. The means and standard deviations of three biological replicates are shown.
(TIFF)

**S3 Fig. TLSs connect cells with both symmetric and asymmetric infection.** Percentage of TLSs connecting WT and AcGFP labelled A549 cells with both asymmetric and symmetric infection status. The means and standard deviations of 3 biological replicates are shown.
(TIFF)\

**S4 Fig. Morphology of budding IAV virions.** Maximum intensity projections of surface HA labelled MDCK cells, at 16 hours post infection with the IAV strains PR8 and Udorn (WT) or the segment 7 reassortant viruses Udorn MPR8 or PR8 MUd. DAPI (blue), HA (yellow). Scale bars = 20 μm.
(TIFF)

**S5 Fig. The cytopathic effect and intercellular spread of IAV is affected by TPCK trypsin and Zanamivir.** (A) Coomassie stained MDCK cells in 12 well plates, 48 hours post infection with the IAV strains PR8 or Udorn under micro-plaque assay conditions. The influence of Zanamivir (0.12 mM) or TPCK trypsin (1 μg/ml) was assessed by cytopathic effect (CPE). Images are representative of two biological replicates. (B) The percentage NP positive foci that form micro-plaques under increasing zanamivir concentrations. Differences between viruses at each concentration were tested for significance by Mann-Whitney test, and differences between concentrations were tested by Kruskal-Wallis test (n.s. $p > 0.05$). The means and standard deviations of three biological replicates are shown. The same data are shown in (C) with each virus normalised to its behaviour in the absence of zanamivir. (D) Mean microplaque area under increasing zanamivir concentrations, with a dashed line showing the approximate area of a single cell. The means and standard deviations of three biological replicates are shown. The significance of differences between viruses was determined by Kruskal-Wallis test (n.s. $p > 0.05$).
(TIFF)

**S6 Fig. Amantadine has a concentration dependent effect on IAV induced cytopathic effect.** Coomassie stained MDCKs within a 12 well plate, 48 hours post infection with either PR8 or Udorn, in the presence (upper panel) and absence (lower panel) of TPCK trypsin at increasing amantadine concentrations. Images are representative of two biological replicates.
(TIFF)

**S7 Fig. Infected cells treated with Z-VAD-fmk show reduced cytological signs of apoptosis, whilst having no antiviral effect.** (A) Representative merged and split channel confocal images of BrightFlu infected (MOI 1.5 PFU/cell) MDCK cells 16 hours post infection, treated with either DMSO or 100 μM Z-VAD-fmk 1 hour post infection. Nuclei (yellow), F-actin (magenta), ZsGreen (green), NP (blue). Scale bar = 20 μm. (B) The percentage of cells imaged that are positive for NP or ZsGreen signal following DMSO or 100 μM Z-VAD-fmk treatment.
(TIFF)

**S8 Fig. Cisplatin treatment activates caspase 3/7 in a dose dependent manner and produces characteristic cytological indicators of apoptosis.** (A) The percentage of cells positive for active caspase 3/7 16 hours post treatment with increasing concentrations of cisplatin. Differences relative to the mock treated control (0 μM) were tested for significance using a Kruskal-Wallis test (n.s. $p > 0.05$, * $p < 0.05$). The mean and SD are shown (n = 3). (B) Representative images of 2x tiled super resolution confocal micrographs of MDCKs treated with 30 μM cisplatin or PBS (mock) for 16 hours. Nuclei (yellow), F-actin (magenta). Scale bar = 20 μm. Images are representative of 3 biological replicates.
(TIFF)

**S9 Fig. TLS inhibiting drugs disrupt the cytoskeleton, with no effect on IAV NP subcellular localisation.** (A) Representative confocal micrographs of MDCKs infected with PR8 at an MOI 0.6, and treated with either DMSO, cytochalasin D (20 μM), taxol (100 μM), and IPA-3 (3 μM) 1 hour post infection. Cells were fixed and immunostained for NP 16 hours post infection. Nuclei (yellow), F-actin (magenta), NP (blue). Scale bar = 20 μm. (B) Average number of TLSs per MDCK cell 16 hours post infection with PR8 at an MOI of 1.5 PFU/cell, either in the presence of DMSO or IPA-3 (3 μM) added 1 hour post infection. Differences in TLS induction was tested for significance by Mann-Whitney test (n.s. $p > 0.05$, *$p < 0.05$). The mean and SD are shown (n = 5).
(TIFF)

**S1 Data. Primary quantitative data.** Collated and indexed quantitative data that underpins all figures of this study. Raw data can be found at the following address: https://doi.org/10.5525/gla.researchdata.1966.
(XLSX)

## Acknowledgments

We would like to thank Colin Nixon for performing the Cryostat tissue sectioning as well as Anna Sims for providing advice, assistance in choice of analysis and helpful discussions. We thank Emma Davies and Matt Turnbull for their advice and the provision of reagents. We would also like to thank Alma Macdonald for granting access to the U.V. lamp.

## Author contributions

**Conceptualization:** Daniel Weir, Edward Hutchinson.

**Data curation:** Daniel Weir.

**Formal analysis:** Daniel Weir.

**Funding acquisition:** Edward Roberts, Edward Hutchinson.

**Investigation:** Daniel Weir, Calum Bentley-Abbot, Jack McCowan, Edward Roberts.

**Methodology:** Daniel Weir, Calum Bentley-Abbot, Colin Loney, Edward Roberts, Edward Hutchinson.

**Supervision:** Edward Hutchinson.

**Validation:** Daniel Weir.

**Visualization:** Daniel Weir, Colin Loney.

**Writing – original draft:** Daniel Weir.

**Writing – review & editing:** Daniel Weir, Edward Hutchinson.

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
