## [Decision Letter · Decision Letter 0]

Dear Mr Weir,

We are pleased to inform you that your manuscript 'Induction of tunnelling nanotube-like structures by influenza A viruses requires the onset of apoptosis' has been provisionally accepted for publication in PLOS Pathogens.

Best regards,

Anice C. Lowen

Academic Editor

PLOS Pathogens

Thomas Hoenen

Section Editor

PLOS Pathogens

Sumita Bhaduri-McIntosh

Editor-in-Chief

PLOS Pathogens

orcid.org/0000-0003-2946-9497

Michael Malim

Editor-in-Chief

PLOS Pathogens

orcid.org/0000-0002-7699-2064

Thank you for your work to address the reviewers' initial concerns. Some additional comments are provided for your consideration but we do not require that you perform additional revisions at this stage. Congratulations on acceptance of your manuscript.

Reviewer Comments (if any, and for reference):

Reviewer's Responses to Questions

**Part I - Summary**

Reviewer #1: The revised manuscript by Weir et al. documents that influenza virus infection induces the formation of tunneling nanotubes that traffic the viral genome between cells. The authors present in vivo (mouse model) data documenting the formation of these structures between columnar epithelial cells of the lower respiratory tract cells, and in vitro analyses documenting that inhibition of apoptosis prevents influenza virus induction of tunneling nanotubes.

Reviewer #2: This revised manuscript by Weir et al. has convincingly addressed some of the points from the previous reviews. They now define TLS clearly, address high and low MOI conditions, and show that the kinetics of caspase activation/apoptosis during infection correlate with the kinetics of induction of TLS. They demonstrate that TLS production does not correlate with the IAV filamentous phenotype. Interestingly, they find that treatment with the apoptosis-inducer cisplatin promotes TLS formation in IAV infected cells at a timepoint post-infection that is otherwise negative (Fig. 7D). They do not address the frequency of the TLS observed in vivo.

Unfortunately, to this reviewer the key point is still unconvincing. The MS does not provide clear evidence that TLS are mediating cell-cell transmission. The authors rely quite a lot on a microplaque assay, but then conclude that it is not useful for evaluation of cell-cell transmission by TLS as the cells are too dense and other mechanisms besides TLSs may be at play . They then use cells plated under low density, to favor TLS-mediated cell-cell transmission, and test various agents to reduce TLS formation. They do see a reduction in transmission by Z-VAD (Fig. 8C). However, other conditions such as cytochalasin D which it appears from the MS strongly reduce TLSs do not reduce transmission. This part of the manuscript struggles to explain what is going on, and overall the logical flow is not convincing of the authors’ thesis.

Reviewer #3: The authors Weir et al have made significant improvements to the manuscript since its first iteration by the addition of several key experiments and expanded discussion. I am satisfied with the manuscript as it stands.

**Part II – Major Issues: Key Experiments Required for Acceptance**

Reviewer #1: The authors have addressed satisfactorily all major issues I raised during the first review.

Reviewer #2: (No Response)

Reviewer #3: (No Response)

**Part III – Minor Issues: Editorial and Data Presentation Modifications**

Reviewer #1: My first review included the suggestion to compare the efficiency by which individual influenza virus infected cells infect their neighbor, which the authors estimated as being 25%, to what occurs with other respiratory viruses. The authors referred to a 2016 analysis of measles virus spread and stated that, to their knowledge, only qualitative evidence for the spread of this virus in primary airway epithelial cells was presented. They may have missed a 2019 publication documenting that measles virus ribonucleocapsids rapidly spread between columnar cells of the respiratory epithelium along F-actin rings (PMID 31772054) apparently with very high (near 100%) efficiency. In addition, a 2015 manuscript contrasts the spread of measles virus in well-differentiated human airway epithelia with that of three other negative strand RNA respiratory viruses (PMID 25926640). Discussing the author's data on influenza virus spread in the context of the mechanisms of spread of other negative strand RNA viruses in the airway epithelium would make the manuscript more generally relevant.

Reviewer #2: (No Response)

Reviewer #3: (No Response)

PLOS authors have the option to publish the peer review history of their article (what does this mean? ). If published, this will include your full peer review and any attached files.

**Do you want your identity to be public for this peer review?** For information about this choice, including consent withdrawal, please see our Privacy Policy .

Reviewer #1: No

Reviewer #2: No

Reviewer #3: No

---

## [Editor Report · Acceptance letter]

Dear Mr Weir,

We are delighted to inform you that your manuscript, "Induction of tunnelling nanotube-like structures by influenza A viruses requires the onset of apoptosis," has been formally accepted for publication in PLOS Pathogens.

Best regards,

Sumita Bhaduri-McIntosh

Editor-in-Chief

PLOS Pathogens

orcid.org/0000-0003-2946-9497

Michael Malim

Editor-in-Chief

PLOS Pathogens

orcid.org/0000-0002-7699-2064